# Overcoming Spurious Solutions in Semi-Dual Neural Optimal Transport: A Smoothing Approach for Learning the Optimal Transport Plan

**Jaemoo Choi** [1]   **Jaewoong Choi** [* 2]   **Dohyun Kwon** [* 3 4]

## Abstract

We address the convergence problem in learning the Optimal Transport (OT) map, where the OT Map refers to a map from one distribution to another while minimizing the transport cost. Semi-dual Neural OT, a widely used approach for learning OT Maps with neural networks, often generates spurious solutions that fail to transfer one distribution to another accurately. We identify a sufficient condition under which the max-min solution of Semi-dual Neural OT recovers the true OT Map. Moreover, to address cases when this sufficient condition is not satisfied, we propose a novel method, OTP, which learns both the OT Map and the Optimal Transport Plan, representing the optimal coupling between two distributions. Under sharp assumptions on the distributions, we prove that our model eliminates the spurious solution issue and correctly solves the OT problem. Our experiments show that the OTP model recovers the optimal transport map where existing methods fail and outperforms current OT-based models in image-to-image translation tasks. Notably, the OTP model can learn stochastic transport maps when deterministic OT Maps do not exist, such as one-to-many tasks like colorization.

## 1. Introduction

Optimal Transport (OT) theory (Villani et al., 2009; Peyré et al., 2019) addresses the problem of finding the cost-optimal transport map that transforms one probability distribution (*source distribution*) into another (*target distribution*). Recently, there has been growing interest in learning

---
[*]Equal contribution   [1]Georgia Institute of Technology [2]Sungkyunkwan University [3]University of Seoul [4]Korea Institute for Advanced Study. Correspondence to: Dohyun Kwon <dh.dohyun.kwon@gmail.com>, Jaewoong Choi <jaewoong-choi@skku.edu>.

*Proceedings of the 42nd International Conference on Machine Learning*, Vancouver, Canada. PMLR 267, 2025. Copyright 2025 by the author(s).

the optimal transport map directly using neural networks. OT has found extensive applications in various machine learning domains by appropriately defining source and target distributions, such as generative modeling (Rout et al., 2022; Choi et al., 2023; 2024b; Choi & Choi, 2024; Lipman et al., 2023), image-to-image translation (Korotin et al., 2023b; Fan et al., 2022), point cloud completion (Lee et al., 2024), and domain adaptation (Flamary et al., 2016). The OT framework is particularly advantageous for unpaired distribution transport tasks, as it relies solely on a predefined cost function to map one distribution to another, eliminating the need for paired data.

Among various approaches, the minimax algorithm, derived from the semi-dual formulation, has been widely investigated (Fan et al., 2022; Rout et al., 2022; Choi et al., 2023; 2024c). Formally, Fan et al. (2022); Rout et al. (2022) established the adversarial algorithm by leveraging the following max-min problem:

$$
\sup_{V} \inf_{T:\mathcal{X}\to\mathcal{Y}} \mathcal{L}(V,T) \quad \text{where} \quad \mathcal{L}(V,T) :=
$$
$$
\int_{\mathcal{X}} c(x,T(x)) - V(T(x))d\mu(x) + \int_{\mathcal{Y}} V(y)d\nu(y). \tag{1}
$$

Here, the probability measures $\mu$ and $\nu$ represent the source and the target distribution, respectively. The function $V : \mathcal{Y} \to \mathbb{R}$ and $T$ approximates a Kantorovich potential (Kantorovich, 1948), and an optimal transport map, respectively. Throughout this paper, we call these approaches the ***Semi-dual Neural OT (SNOT)***.

When the optimal potential $V^\star$ and the transport map $T^\star$ exist, it is well-known that

$$
T^\star \in \arg\min_{T} \mathcal{L}(V^\star, T). \tag{2}
$$

as shown in Rout et al. (2022); Fan et al. (2022). Thus, the pair $(V^\star, T^\star)$ is the solution to this max-min problem. However, a critical challenge arises: not all solutions of Eq.1 correspond to the optimal potential and transport map pair. In other words, even the optimal solution in the SNOT framework may not recover the correct optimal transport map. We refer to this challenge as the ***spurious solution problem***.

In this paper, we analyze this fundamental issue of the spurious solution problem in existing SNOT frameworks. Specif-

ically, we identify a sufficient condition on the source distribution $\mu$ that prevents the spurious solution problem. The key condition is that the source distribution should not place positive mass on measurable sets with Hausdorff dimension $\leq d - 1$ (see Thm 3.1). To the best of our knowledge, our work offers the first theoretical analysis of the sufficient condition under which the max-min solution of the SNOT framework can correctly learn the OT Map. Prior works were limited to the saddle point solution (Fan et al., 2023) or addressed a specific form of a different OT problem (weak OT) (Korotin et al., 2023b;a) (see Appendix D for the related works on spurious solution issues). Additionally, we comprehensively explore various failure cases when this condition is not satisfied.

Building on this condition, we develop a novel algorithm that ensures the learning of an optimal transport plan. We refer to our model as the ***Optimal Transport Plan model (OTP)***. Our method involves smoothing the source distribution $\mu_\epsilon$, so that the Neural OT models recover the correct optimal transport plan. Then, we gradually modify $\mu_\epsilon$ back to the original $\mu$ leveraging the convergence property. Our extensive experiments show that our OTP model accurately learns the optimal transport plan. Moreover, our model outperforms various (entropic) Neural OT models in diverse image-to-image translation tasks. Our contributions can be summarized as follows:

- Our work is the first to identify a sufficient condition under which the max-min solution of existing SNOT recovers the true OT Map.

- We demonstrate diverse failure cases that occur when this sufficient condition is not satisfied.

- We propose a new algorithm that guarantees the learning of the optimal transport plan.

- Our experiments show that our model successfully recovers the correct OT Plan in failure cases where existing models fail.

**Notations and Assumptions**   Let $(\mathcal{X}, \mu)$ and $(\mathcal{Y}, \nu)$ be Polish spaces where $\mathcal{X}$ and $\mathcal{Y}$ are closures of connected open sets in $\mathbb{R}^d$. We regard $\mu$ and $\nu$ as the source and target distributions. Unless otherwise described, we consider $\mathcal{X} = \mathcal{Y} = \mathbb{R}^d$ and the quadratic transport cost $c : \mathcal{X} \times \mathcal{Y} \rightarrow \mathbb{R}$, $c(x, y) = \alpha\|x - y\|^2$ for a given positive constant $\alpha$. For a measurable map $T$, $T_{\#}\mu$ represents the pushforward distribution of $\mu$. $\Pi(\mu, \nu)$ denotes the set of joint probability distributions on $\mathcal{X} \times \mathcal{Y}$ whose marginals are $\mu$ and $\nu$, respectively. Moreover, we denote $W_2(\cdot, \cdot)$ as the 2-Wasserstein distance of two distributions.

## 2. Background

In this section, we present a brief overview of Optimal Transport theory (Villani et al., 2009; Santambrogio, 2015), and neural network approaches for learning optimal transport

maps. In particular, we focus on approaches that leverage the semi-dual formulation (Rout et al., 2022; Fan et al., 2022).

**Optimal Transport**   The Optimal Transport (OT) problem investigates transport maps that connect the source distribution $\mu$ and the target distribution $\nu$ (Villani et al., 2009; Santambrogio, 2015). The *optimal transport map (OT Map or Monge Map)* is defined as the minimizer of a given cost function among all transport maps between $\mu$ and $\nu$. Formally, Monge (1781) introduced the OT problem with a deterministic transport map $T$ as follows:

$$\mathcal{T}(\mu, \nu) := \inf_{T_{\#}\mu=\nu} \left[ \int_{\mathcal{X}} c(x, T(x))d\mu(x) \right]. \quad (3)$$

Note that the condition $T_{\#}\mu = \nu$ indicates that the transport map $T$ transforms $\mu$ to $\nu$, where $T_{\#}\mu$ denotes the pushforward distribution of $\mu$ under $T$. However, the Monge OT problem is non-convex, and the existence of minimizer, i.e., the optimal transport map $T^\star$, is not always guaranteed depending on the assumption of $\mu$ and $\nu$ (Sec. 3.2.2).

To address this existence issue, Kantorovich (1948) proposed the following convex formulation of the OT problem:

$$C(\mu, \nu) := \inf_{\pi \in \Pi(\mu, \nu)} \left[ \int_{\mathcal{X} \times \mathcal{Y}} c(x, y)d\pi(x, y) \right], \quad (4)$$

We refer to the joint probability distribution $\pi \in \Pi(\mu, \nu)$ as the *transport plan* between $\mu$ and $\nu$. Unlike the Monge OT problem, the optimal transport plan (OT Plan) $\pi^\star$ is guaranteed to exist under mild assumptions on $(\mathcal{X}, \mu)$ and $(\mathcal{Y}, \nu)$ and the cost function $c$ (Villani et al., 2009). Intuitively, while the Monge OT (Eq. 3) covers only the deterministic transport map $y = T(x)$, the Kantorovich OT problem (Eq. 4) can represent stochastic transport map via the conditional distribution $\pi(y|x)$ for each $x \sim \mu$. When the optimal transport map $T^\star$ exists, the optimal transport plan also reduces to this deterministic transport map, i.e., $\pi^\star = (Id \times T^\star)_{\#}\mu$.

**Semi-dual Neural OT**   The goal of neural optimal transport (Neural OT) models is to learn the OT Map between $\mu$ and $\nu$ using neural networks. The semi-dual formulation of the OT problem is widely leveraged for learning OT Maps (Rout et al., 2022; Fan et al., 2022; Choi et al., 2023; Makkuva et al., 2020).

The semi-dual formulation of the OT problem is given as follows: For a general cost function $c(\cdot, \cdot)$ that is lower semicontinuous and bounded below, the Kantorovich OT problem (Eq. 4) has the following *semi-dual form* (Villani et al., 2009, Thm. 5.10), (Santambrogio, 2015, Prop. 1.11):

$$S(\mu, \nu) := \sup_{V \in S_c} \left[ \int_{\mathcal{X}} V^c(x)d\mu(x) + \int_{\mathcal{Y}} V(y)d\nu(y) \right], \quad (5)$$

where $S_c$ denotes the collection of $c$-concave functions $\psi : \mathcal{Y} \rightarrow \mathbb{R}$ and $V^c$ denotes the $c$-transform of $V$, i.e.,

$$V^c(x) = \inf_{y \in \mathcal{Y}} \left[ c(x,y) - V(y) \right]. \qquad (6)$$

The SNOT approaches utilize this semi-dual form (Eq. 5) for learning the OT Map $T^\star$ (Rout et al., 2022; Fan et al., 2022; Makkuva et al., 2020). This formulation leads to a max-min optimization problem, similar to GANs (Goodfellow et al., 2020). Specifically, these models parametrize the transport map $T_\theta : \mathcal{X} \to \mathcal{Y}$ and the potential $V_\phi$ as follows:

$$T_\theta : x \mapsto \arg \min_{y \in \mathcal{Y}} \left[ c(x,y) - V_\phi(y) \right] \qquad (7)$$

$$\Leftrightarrow \quad V_\phi^c(x) = c(x, T_\theta(x)) - V_\phi(T_\theta(x)). \qquad (8)$$

Note that $T_\theta$-parametrization (Eq. 7) implies that the $c$-transform $V_\phi^c$ can be expressed with the transport map $T_\theta$ and the potential $V_\phi$, as shown in Eq. 8. From this, the SNOT models derive the following optimization problem $\mathcal{L}_{V_\phi, T_\theta}$:

$$\sup_{V_\phi \in S_c} \inf_{T_\theta : \mathcal{X} \to \mathcal{Y}} \mathcal{L}(V_\phi, T_\theta) \quad \text{where} \quad \mathcal{L}(V, T) :=$$
$$\int_{\mathcal{X}} c(x, T(x)) - V(T(x)) \, d\mu(x) + \int_{\mathcal{Y}} V(y) d\nu(y). \qquad (9)$$

Intuitively, $T_\theta$ and $V_\phi$ serve similar roles to the generator and the discriminator in GANs. However, the OT Map $T_\theta$ is additionally trained to minimize the transport cost $c(x, T_\theta(x))$, while GANs focus solely on learning the target distribution $T_{\#}\mu = \nu$ (Arjovsky et al., 2017; Gulrajani et al., 2017).

## 3. Analytical Results for Semi-dual Neural OT

A critical limitation of existing SNOT approaches is that the max-min solution $(V^\dagger, T^\dagger)$ of Eq. 9 may include not only the desired OT Map but also other **spurious solutions** (Rout et al., 2022). Formally, if the OT Map $T^\star$ exists, the optimal potential $V^\star$ and the OT Map $T^\star$ become a max-min solution (Eq. 2). However, not all max-min solutions correspond to the true optimal potential and transport map, i.e., $\{(V^\star, T^\star)\} \subsetneq \{(V^\dagger, T^\dagger)\}$. In particular, even $T^\dagger \# \mu = \nu$ does not hold in general (see Fig. 1), which means that $T^\dagger$ is not a valid transport map from $\mu$ to $\nu$ as in (Eq. 3). It is worth noting that any max-min solution satisfying $T^\dagger \# \mu = \nu$ is guaranteed to be the OT map (Fan et al., 2023, Thm 3).

We first investigate sufficient conditions to prevent spurious solution issues (Sec. 3.1), and present a comprehensive failure case analysis of the SNOT approach (Sec. 3.2). Based on this, later in Sec. 4, we propose a method for learning an accurate Neural OT model that avoids such spurious solutions.

### 3.1. Sufficient Conditions for Ensuring Convergence of Semi-dual Neural OT

We provide sufficient conditions on the source distribution $\mu$ and the target distribution $\nu$ to ensure a unique minimizer for the $T_\theta$-parametrization (Eq. 7). This enables the SNOT objective to accurately recover the optimal transport plan.

**Theorem 3.1.** *Let $\mu \in \mathcal{P}_2(\mathcal{X}), \nu \in \mathcal{P}_2(\mathcal{Y})$ and $c(x,y) = \frac{1}{2}\|x - y\|^2$. Assume that $\mu$ does not give mass to the measurable sets of Hausdorff dimension at most $d - 1$ dimension.*

*(1) Then, there exists a unique OT Map $T^\star$ in (Eq. 3) and a (possibly non-unique) Kantorovich potential $V^\star \in S_c$ in (Eq. 5).*

*(2) For the Kantorovich potential $V^\star \in S_c$, a solution of the minimization problem,*
$$\mathcal{D}_x := \arg \min_{y \in \mathcal{Y}} \left[ c(x,y) - V^\star(y) \right], \qquad (10)$$

*is uniquely determined $\mu$-a.s., i.e. $\mathcal{D}_x = \{y_x\}$ for $\mu$-a.s $x \in \mathcal{X}$. In particular, a map $x \mapsto y_x \in \mathcal{D}_x$ is a unique OT Map $T^\star$ $\mu$-a.s..*

Here, $\mathcal{D}_x$ corresponds to the $T_\theta$-parametrization in the SNOT framework. Therefore, the uniqueness of $\mathcal{D}_x$ for $V^\star$ implies that $T_\theta$-parametrization is fully characterized. **Thm. 3.1 shows that the assumption on $\mu$, not on $\nu$, is enough to eliminate the ambiguity in mapping each $x$ to $T_\theta(x)$ and this mapping corresponds to the OT Map.** Furthermore, the extension of Thm. 3.1 to a general cost function is discussed in Thm. A.6 from Appendix A.1.

Note that Thm. 3.1 is sufficient for addressing the spurious solution problem. For the sake of completeness, we also present a sufficient condition where the SNOT framework admits **a unique max-min solution that corresponds to the correct OT Map** (Cor. 3.3). In this case, the additional assumptions on $\nu$ is also required. Here, we used the fact that the absolutely continuous measures with respect to the Lesbesgue measure satisfy the condition in Thm. 3.1.

**Theorem 3.2.** *Suppose $\mathcal{Y} \subset \mathbb{R}^d$ is a closure of a bounded open set. If $\nu$ has a positive density almost everywhere with respect to the Lebesgue measure on $\mathcal{Y}$, then there exists unique Kantorovich potential $V^\star \in S_c$ up to constant.*

Cor. 3.3 is derived by combining Thm. 3.1 with the uniqueness of the optimal potential $V^\star$ in Thm. 3.2.

**Corollary 3.3.** *Suppose $\mathcal{Y} \subset \mathbb{R}^d$ is a closure of a bounded open set. Suppose $\mu \in \mathcal{P}_2(\mathcal{X})$ and $\nu \in \mathcal{P}_2(\mathcal{Y})$ are absolutely continuous distributions that have positive density functions on their domain. Then, the solution $(V^\star, T^\star)$ of equation (9) is unique. In other words, $V^\star \in S_c$ is unique up to constant, and $T^\star$ is a deterministic OT Map.*

### 3.2. Failure Cases When Our Condition Is Not Met

In Thm 3.1, it is crucial to assume that $\mu$ does not give mass to the measurable sets of Hausdorff dimension at most $d - 1$ dimension. Without this assumption, SNOT may fail even when the deterministic OT Map $T^\star$ uniquely exists. Specifically, the failure cases discussed in this section refer to scenarios where $(V^\dagger, T^\dagger)$ is a max-min solution of Eq. 1 but does not correspond to the OT Map $T^\star$ (Eq. 3), i.e., $(Id, T^\dagger)_{\#}\mu$ fails to represent the OT Plan $\pi^\star$ (Eq. 4).

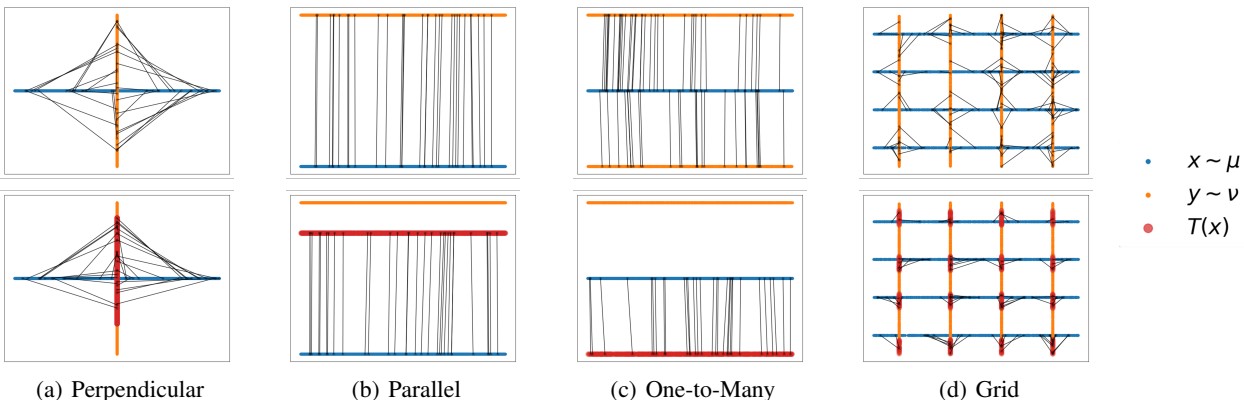

(a) Perpendicular      (b) Parallel      (c) One-to-Many      (d) Grid

*Figure 1.* **Visualization of failure cases** by comparing the Optimal Transport map (**1st row**) and the max-min solution (**2nd row**) of Semi-dual Neural OT in the failure cases. The source data $x \sim \mu$, target data $y \sim \nu$, and generated data $T(x)$ are represented in Blue, Orange, and Red. The max-min solution fails to recover the correct OT Map.

### 3.2.1. DISCREPANCY BETWEEN A MAX-MIN SOLUTION AND THE DETERMINISTIC OT MAP

We first focus on the Monge OT problem (Eq. 3), where the deterministic OT Map $T^\star$ exists. Specifically, we investigate source and target distribution pairs where $T^\star$ exists, but the max-min solution $T^\dagger$ of the SNOT objective (Eq. 9) fails to recover this optimal solution. Here, we provide two examples, depending on the uniqueness of $T^\star$.

**Example 1. [When $T^\star$ exists but is not unique]** First, we introduce a case where multiple optimal solutions $T^\star$ exist for the Monge OT problem. Assume that the source and target distributions are uniformly supported on $A = [-1, 1] \times \{0\}$ and $B = \{0\} \times [-1, 1]$, respectively (Fig. 1(a)). In this case, any transport map $T$ satisfying $T_{\#}\mu = \nu$ becomes an optimal transport map for the quadratic cost function. Formally, note that for any transport map $T$, the following holds:

$$\int_{\mathcal{X}} c(x, T(x)) d\mu(x) = \frac{1}{2} \int_{\mathcal{X}} \|x\|^2 d\mu(x)$$
$$- \int_{\mathcal{X}} \langle x, T(x) \rangle \, d\mu(x) + \frac{1}{2} \int_{\mathcal{Y}} \|y\|^2 d\nu(y) = \frac{1}{3}. \quad (11)$$

The first equality follows from $T_{\#}\mu = \nu$ and $\langle x, T(x) \rangle = 0$ for all $x$ because $A \perp B$. Since every transport map achieves the same transport cost, any transport map becomes an optimal transport map $T^\star$.

Then, we prove that $T^\dagger$ does not correspond to $T^\star$. Specifically, we show that $V^\star(y) \in S_c$[1] is the Kantorovich potential (Eq. 5) and that $T^\dagger$ is not guaranteed to generate the

---

[1]Note that $V^\star(y)$ is $c$-concave function since $\frac{1}{2}\|y\|^2 - V^\star(y)$ is convex and lower semicontinuous (Thm 1.21, (Santambrogio, 2015)).

target distribution.

$$V^\star(y) := \begin{cases} \frac{1}{2}\|y\|^2 & \text{if } y \in B \\ -\infty & \text{otherwise.} \end{cases} \quad (12)$$

By substituting $V^\star$ into $V$, the inner problem of SNOT (Eq. 9) can be expressed as follows.

$$\inf_{T:\mathbb{R}^2 \to \mathbb{R}^2} \int_{\mathcal{X}} \left( \frac{1}{2}\|x - T(x)\|^2 - V^\star(T(x)) \right) d\mu(x) \quad (13)$$
$$= \inf_{T \text{ s.t. supp}(T_{\#}\mu) \subset B} \int_{\mathcal{X}} \left( \frac{1}{2}\|x - T(x)\|^2 - \frac{1}{2}\|T(x)\|^2 \right) d\mu(x)$$
$$= \inf_{T \text{ s.t. supp}(T_{\#}\mu) \subset B} \int_{\mathcal{X}} \frac{1}{2}\|x\|^2 - \langle x, T(x) \rangle d\mu(x)$$
$$= \int_{\mathcal{X}} \frac{1}{2}\|x\|^2 d\mu(x).$$

Note that $T(x) \in B$ for $\mu$-a.s. in $x$; otherwise, the value of Eq. 13 becomes infinite. Therefore, $\langle x, T(x) \rangle = 0$ and $V^\star(T(x)) = \frac{1}{2}\|T(x)\|^2$. Then, Eq. 9 becomes as follows

$$\int_{\mathcal{X}} \frac{1}{2}\|x\|^2 d\mu(x) + \int_{\mathcal{Y}} \frac{1}{2}\|y\|^2 d\nu(y) = \frac{1}{3}. \quad (14)$$

Since $V^\star$ attains the same value of $\mathcal{L}_{V_\phi, T_\theta}$ as $T^\star$ in Eq. 11, $V^\star$ is the optimal potential. Furthermore, by comparing Eq. 13 with $T_\theta$-parametrization (Eq. 7), we can easily observe that any measurable map $T_\theta : A \to B$ can be a max-min solution of SNOT. In other words, there is no constraint ensuring that $T_{\theta\#}\mu = \nu$. For example, $T_\theta(x) = (0, 0)$ for $\forall x \in \mathcal{X}$ is also a valid max-min solution. This means that the existing SNOT models cannot learn the optimal transport map between these two distributions (Fig. 3(a)).

**Example 2. [When unique $T^\star$ exists]** Here, we present another failure case when there is a unique optimal transport map $T^\star$. Assume that the source and target distributions

are uniformly distributed over $A = [-1, 1] \times \{0\}$ and $B = [-1, 1] \times \{1\}$, respectively (Fig. 1(b)). In this setup, the unique $T^\star$ is given by:

$$T^\star(x) := (x_1, 1) \quad \text{for } x = (x_1, 0) \in \mathcal{X}. \quad (15)$$

Thus, $\mathcal{T}(\mu, \nu) = \frac{1}{2}$. Similar to Example 1, we show that $V^\star(y) = \frac{1}{2}\|y_2\|^2 \in S_c$ with $y = (y_1, y_2)$ is the optimal Kantorovich potential and analyze the max-min solution of Eq. 9. For this $V^\star$, the inner problem of SNOT can be computed as follows:

$$\inf_T \int_\mathcal{X} \frac{1}{2}\|x_1 - T(x)_1\|^2 d\mu(x) + \int_\mathcal{Y} \frac{1}{2}\|y\|^2 d\nu(y) = \frac{1}{2}. \quad (16)$$

Because Eq. 16 achieves the same value as $\mathcal{T}(\mu, \nu)$, $V^\star$ is the optimal potential. For this $V^\star$, any transport map $T((x_1, x_2)) := (x_1, a)$ for any $a \in \mathbb{R}$ for each $(x_1, x_2) \in \mathcal{X}$ becomes a max-min solution of the SNOT. In this case, the existing approach fails to even characterize the correct support of the target distribution $\nu$.

### 3.2.2. DISCREPANCY BETWEEN A MAX-MIN SOLUTION AND THE STOCHASTIC OT MAP

The standard SNOT parametrizes the transport map with a deterministic function $T_\theta$ (Eq. 7). When no deterministic OT Map $T^\star$ exists but only an OT Plan $\pi^\star$ exists (Eq. 4), it is clear that the SNOT cannot accurately represent the stochastic OT Map (OT Plan).

**Example 3. [When only $\pi^\star$ exists]** Suppose the source and target distributions are uniform on $A = [0, 1] \times \{0\}$ and $B = [0, 1] \times \{1\} \cup [0, 1] \times \{-1\}$, respectively (Fig. 1(c)). In this case, it is clear that the OT Plan $\pi^\star$ is given as follows:

$$\pi^\star(y|x) = \frac{1}{2}\delta_{(x_1, 1)} + \frac{1}{2}\delta_{(x_1, -1)} \text{ where } x = (x_1, x_2). \quad (17)$$

The OT Plan $\pi^\star(y|x)$ moves each $x$ vertically either up or down with probability $\frac{1}{2}$, without incurring additional cost from horizontal movement. Then, we show that $V^\star(y) = \frac{1}{2}\|y_2\|^2 \in S_c$ with $y = (y_1, y_2)$ is the optimal potential. The $(V^\star)^c$ and $V^\star$ can be computed for $\mu$ and $\nu$ as follows:

$$(V^\star)^c(x) = \inf_{y \in \mathcal{Y}}(c(x, y) - V^\star(y)) = \inf_{y_1} \frac{1}{2}\|x_1 - y_1\|^2 = 0, \quad (18)$$

$V^\star(y) = \frac{1}{2}\|y_2\|^2 = \frac{1}{2}$ for $\forall y \in \mathcal{Y}$. By comparing the $C$ for the optimal transport plan (Eq. 17) and the semi-dual form $S$ for $V^\star$, we can easily verify that $V^\star$ is the optimal Kantorovich potential.

Then, from Eq. 18, we can see that $T_1$ and $T_2$ are the two possible solutions for the $T$-parametrization (Eq. 7) in the SNOT for $V^\star$, $T_1(x) := (x_1, 1)$ and $T_2(x) := (x_1, -1)$. for $x = (x_1, x_2) \in \mathcal{X}$. These two candidates $T_1, T_2$ only characterize a subset of the support of $\pi^\star(y|x)$. Therefore, our deterministic $T_\theta$ cannot learn the stochastic $\pi^\star(y|x)$.

---

**Algorithm 1** Training algorithm of OTP

---

**Require:** Source distribution $\mu$ and the target distribution $\nu$; OT Map network $T_\theta$ and potential network $V_\phi$; Total number of iteration $K$; Number of inner-loop iterations $K_T$; Decreasing sequence of noise levels $\{\epsilon_k\}_{k=1}^K$.

1: **for** $k = 0, 1, 2, \ldots, K$ **do**
2:     Sample a batch $x \sim \mu, y \sim \nu, z \sim \mathcal{N}(\mathbf{0}, \mathbf{I})$.
3:     $\tilde{x} \leftarrow x + \sqrt{\epsilon_k}z$ or $\tilde{x} \leftarrow \sqrt{1 - \epsilon_k}x + \sqrt{\epsilon_k}z$.
4:     Update $\phi$ to maximize $\mathcal{L}_\phi = -V_\phi(T_\theta(\tilde{x})) + V_\phi(y)$.
5:     **for** $j = 0, 1, \ldots, K_T$ **do**
6:         Sample a batch $x \sim \mu, z \sim \mathcal{N}(\mathbf{0}, \mathbf{I})$.
7:         $\tilde{x} \leftarrow x + \sqrt{\epsilon_k}z$ or $\tilde{x} \leftarrow \sqrt{1 - \epsilon_k}x + \sqrt{\epsilon_k}z$.
8:         $\mathcal{L}_\theta = c(\tilde{x}, T_\theta(\tilde{x})) - V_\phi(T_\theta(\tilde{x})) + V_\phi(y)$.
9:         Update $\theta$ to minimize $\mathcal{L}_\theta$.
10:     **end for**
11: **end for**

---

**Stochastic Parametrization of OT Map** In practice, a stochastic parametrization of $T_\theta(x, z)$ is often adopted to improve performance in the SNOT models (Korotin et al., 2023b; Choi et al., 2023). This stochastic parametrization $T_\theta(x, z)$ introduces an additional noise variable $z \sim N(0, I)$:

$$T_\theta(x, z) \in \arg\min_{y \in \mathcal{Y}}\{c(x, y) - V^\star(y)\}, \quad (19)$$

$(x, z) \sim \mu \times \mathcal{N}(0, I)$ a.s.. As a result, each $x$ is transported to multiple $T(x, z)$ values depending on $z$. **We point out that even a stochastic parametrization, such as $T_\theta(x, z)$ with a noise variable $z \sim N(0, I)$, cannot address this limitation.** For the formal statement, see Appendix B.

**Proposition 3.4** (Informal)**.** *Assume that the stochastic parametrization of $T_\theta(x, z)$ is ideally trained as in equation (19) for $(\mu, \mathcal{N})$-a.s. $\mathcal{D}_x$ in Eq. 10 may not uniquely determined and $T_\theta(x, z)$ may contain spurious solutions.*

## 4. Method

In Sec. 3.2, we analyzed the sufficient condition to prevent failures in the existing SNOT framework. Building on this analysis, we propose a novel method for learning the OT Plan, called the *Optimal Transport Plan (OTP) model*, which is effective even when the conditions are not satisfied.

### 4.1. Proposed Method

**Our goal is to learn the OT Plan $\pi^\star$ (Eq. 4) between the source distribution $\mu$ and the target distribution $\nu$.** Note that the sufficient condition in Thm. 3.1 is an inherent property of $\mu$. When this condition is not satisfied, the existence of OT Map $T^\star$ is not guaranteed, and only $\pi^\star$ exists. In this regard, our OTP model serves as a natural generalization of existing SNOT models.

Our method consists of two steps: First, we introduce a

smoothed version of the source distribution $\mu_\epsilon$. $\mu_\epsilon$ is constructed to satisfy the sufficient conditions from Thm. 3.1. As a result, the SNOT between $\mu_\epsilon$ and $\nu$ recovers the correct OT Plan $\pi_\epsilon^\star$ between them. Second, we gradually adjust $\mu_\epsilon$ back to the original source measure $\mu$. This approach allows our method to learn the correct optimal transport plan, even in cases where the existing SNOT framework fails.

**OTP Model** As a practical implementation of the high-level scheme described above, we propose a new method for learning the OT Plan $\pi^\star$ from $\mu$ to $\nu$, called *Optimal Transport Plan (OTP)* model. This method is based on Thm. 3.1 and Thm. 4.1, which require the following two conditions on the smoothed measure $\mu_\epsilon$:

(c1) $\mu_\epsilon$ does not give mass to the measurable sets of Hausdorff dimension at most $d-1$ dimension (Thm. 3.1).

(c2) $\mu_{\epsilon_k}$ weakly converges to $\mu$ as $k \to \infty$ (Thm. 4.1).

For simplicity, we consider the absolute continuity condition on $\mu_\epsilon$ as we did in Cor. 3.3. Motivated by diffusion models (Ho et al., 2020; Song et al., 2021), we consider **two options for the smoothing distribution**: (1) Gaussian convolution $\mu_{\epsilon_k} = \mu * \mathcal{N}(0, \epsilon_k I)$ and (2) Variance-preserving convolution $\mu_{\epsilon_k} = \left(\sqrt{1-\epsilon_k}Id\right)_{\#} \mu * \mathcal{N}(0, \epsilon_k I)$ with a predefined noise level $\epsilon_k \searrow 0$. For noise-level scheduling, we follow Song et al. (2021). Note that both of these smoothing distributions satisfy conditions (c1) and (c2). Specifically, for Gaussian convolution, for any $\mu \in \mathcal{P}_2(\mathbb{R}^d)$, (c1) $\mu_\epsilon$ is absolutely continuous with respect to the Lebesgue measure and has positive density on $\mathbb{R}^d$. Moreover, (c2) as $\epsilon \to 0$, $\mu_\epsilon \rightharpoonup \mu$. A similar argument works for the Variance-preserving convolution case. Then, we apply the SNOT framework to the smoothed measure $\mu_{\epsilon_k}$ and the target measure $\nu$. The learning objective is given as follows:

$$\mathcal{L}_{V_\phi, T_\theta}^k = \sup_{V_\phi} \left[ \int_\mathcal{X} \inf_{T_\theta} \left[ c\left(x, T_\theta(x)\right) - V_\phi\left(T_\theta(x)\right) \right] d\mu_{\epsilon_k}(x) \right. \\ \left. + \int_\mathcal{X} V_\phi(y) d\nu(y) \right]. \quad (20)$$

Then, we gradually decrease the noise level $\{\epsilon_k\}_{k=1}^K$ throughout training. The two conditions on $\mu_{\epsilon_k}$, i.e., (c1) and (c2), offer the following guarantees. First, for each noise level $\epsilon_k$, the max-min solution of $\mathcal{L}_{V_\phi, T_\theta}^k$ recovers the optimal transport map $T_k^\star$ and the Kantorovich potential $V_k^\star$. Second, as $k \to \infty$, i.e., $\epsilon_k \searrow 0$, the optimal transport plan $\pi_k^\star = (Id, T_k^\star)_{\#} \mu_{\epsilon_k}$ converges (up to a subsequence) to $\pi^\star$. Thm. 4.1 follows from combining Thm. 3.1 and Villani et al. (2009). See Appendix C for proof.

**Theorem 4.1.** *Let $\{\mu_{\epsilon_k}\}_{k\in\mathbb{N}}$ be a sequence of absolutely continuous probability measures, and $T_k^\star$ be the OT map from $\mu_{\epsilon_k}$ to $\mu$. If $\mu_{\epsilon_k}$ weakly converges to $\mu$ as $k \to \infty$, then $\pi_k^\star = (Id, T_k^\star)_{\#} \mu_{\epsilon_k}$ weakly converges to the OT plan $\pi^\star$ between $\mu$ and $\nu$, along a subsequence. Consequently,*

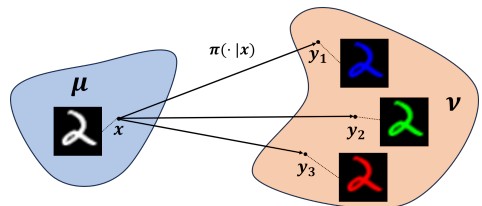

*Figure 2.* **Example of a stochastic transport map (OT Plan) task**, e.g., MNIST-to-CMNIST colorization.

$\pi_k^\star$ *from our OTP model with either convolution above also weakly converges to $\pi^\star$, along a subsequence.*

In this way, we can learn the optimal transport plan $\pi^\star$ between $\mu$ and $\nu$ without falling into the spurious solutions of the max-min learning objective (Eq. 9). While the convergence theorem only guarantees convergence up to a subsequence (Thm. 4.1), our method exhibits decent convergence to $\pi^\star$ in practice (Sec. 5). Specifically, our training algorithm progressively finetunes the transport network $T_\theta$ and the potential network $V_\phi$ by adjusting the smoothing level. As a result, the subsequence convergence does not pose any issues.

**Importance of OT Plan in Neural OT** Our OTP model is for learning the OT Plan, i.e., the stochastic transport map. In fact, OT Plans are not only a theoretical generalization of deterministic OT Maps, but are also inherently more suitable for various real-world machine learning applications. For instance, in image-to-image translation tasks, stochastic OT Plans can effectively model the diversity of plausible outputs. Similarly, in inverse problems such as colorization or image inpainting, stochastic OT Plans are also highly desirable because these tasks inherently involve multiple possible solutions. In Sec 5, our experiments show that our OTP model is effective in handling the stochastic transport map application in the MNIST-to-CMNIST image translation task (Fig. 2).

**Algorithm** We present our training algorithm for OTP (Algorithm 1). For each $\epsilon_k$, we alternatively update the adversarial learning objective $\mathcal{L}_{V_\phi, T_\theta}^k$ between the potential function $V_\phi$ and the transport map $T_\theta$, similar to the GAN framework (Goodfellow et al., 2020). Note that the smooth source measure $\mu_{\epsilon_k}$ corresponds to the probability distribution of the sum of the clean source measure $\mu$ (or the scaled source measure $\left(\sqrt{1-\epsilon_k}Id\right)_{\#} \mu$) and the Gaussian noise $\mathcal{N}(0, \epsilon_k I)$. Therefore, we can easily sample $x_{\epsilon_k} \sim \mu_\epsilon$, as follows (Line 3):

$$x_{\epsilon_k} = x + \sqrt{\epsilon_k} z \sim \mu * \mathcal{N}(0, \epsilon_k I) \quad \text{or} \quad x_{\epsilon_k} = \sqrt{1-\epsilon_k} x + \sqrt{\epsilon_k},$$
$$(21)$$

where $x \sim \mu$ and $z \sim \mathcal{N}(0, I)$. In practice, decreasing the noise level until a small positive constant $\epsilon_{min} > 0$ provided better performance and training stability, compared to reducing the noise level to exactly zero. For a fair comparison, we

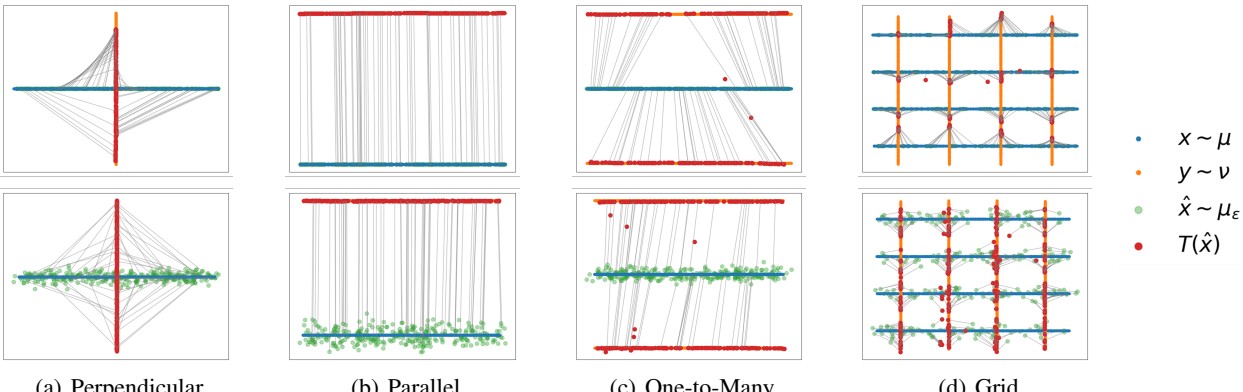

|      |      |      |      |
| :--: | :--: | :--: | :--: |
| (a) Perpendicular | (b) Parallel | (c) One-to-Many | (d) Grid |

*Figure 3.* **Qualitative comparison between OTM (1st row) and our model (2nd row) on failure cases** in Sec 3.2. The noised source sample $\tilde{x}$ in Alg 1 is denoted in Green. While OTM falls into spurious solutions and fails to generate the target distribution correctly, our OTP model successfully learns the OT Plan.

compared the composition of the noising and transport map $x \mapsto x_{\epsilon_{min}} \mapsto T_\theta(x_{\epsilon_{min}})$, with the ground-truth optimal transport map $x \mapsto T^\star(x)$ in the experiments (Sec. 5).

## 5. Experiments

In this section, we evaluate our OTP model from the following perspectives. In Sec. 5.1, we evaluate whether OTP successfully learns the optimal transport plan. In Sec. 5.2, we demonstrate the scalability of OTP by assessing it on the image-to-image translation task. For implementation details of experiments, please refer to Appendix E.

### 5.1. OT Plan Evaluation on Failure Cases

First, **we assess whether our model accurately learns the optimal transport plan $\pi^\star$ between the source distribution $\mu$ and the target distribution $\nu$ in failure cases outlined in Sec 3.2.** The evaluation is conducted in two settings: (1) Qualitative comparison in 2D cases and (2) Quantitative comparison in high-dimensional cases. In each setting, our OTP model is compared against the existing SNOT framework (Eq. 9).

**Qualitative Comparison** In Sec. 3.2, we presented various examples where the existing SNOT framework may fail to learn the OT Map (or Plan). Here, we demonstrate that the existing approaches indeed encounter these failures, while OTP successfully learns the correct OT Map. As a baseline, we compare our method against the standard OTM with a deterministic transport map, i.e., $T_\theta(x)$.

Fig. 3 presents qualitative results on four datasets: Perpendicular (Ex.1), Parallel (Ex.2), One-to-Many (Ex.3), and Grid. The first row shows the vanilla OTM results and the second row exhibit our OTP results. Note that our OTP decreases the noise level until $\sigma = \epsilon_{min} > 0$ (Sec. 4). Hence, the noised source samples $\tilde{x}$ in Alg 1 (Green in Fig 3) are

*Table 1.* **Quantitative comparison of numerical accuracy** on synthetic datasets. Each model is evaluated using two metrics: transport cost error $D_{cost}(\downarrow)$ and target distribution error $D_{target}(\downarrow)$.

| Dimension | Model | Perpendicular | | One-to-Many | |
| :---: | :---: | :---: | :---: | :---: | :---: |
| | | $D_{cost}$ | $D_{target}$ | $D_{cost}$ | $D_{target}$ |
| $d=2$ | OTM | 0.038 | 0.008 | 0.069 | 0.100 |
| | OTM-s | **0.007** | 0.018 | 0.350 | **0.032** |
| | Ours | 0.019 | **0.007** | **0.002** | 0.110 |
| $d=4$ | OTM | 0.043 | 0.039 | 0.100 | 0.090 |
| | OTM-s | **0.033** | 0.065 | **0.010** | **0.038** |
| | Ours | 0.089 | **0.009** | 0.033 | 0.094 |
| $d=16$ | OTM | 0.160 | 4.97 | 71.28 | 73.23 |
| | OTM-s | 0.061 | 4.85 | 97.49 | 99.57 |
| | Ours | **0.058** | **0.59** | **0.06** | **0.65** |
| $d=64$ | OTM | 2.13 | 19.37 | 21.92 | 32.94 |
| | OTM-s | 2.74 | 18.79 | 0.20 | 12.21 |
| | Ours | **0.97** | **10.09** | **0.14** | **9.98** |
| $d=256$ | OTM | 10.98 | 84.91 | **0.04** | 62.02 |
| | OTM-s | 16.45 | 81.68 | 0.25 | 61.87 |
| | Ours | **5.05** | **63.36** | 0.33 | **61.27** |

transported to the target measure $\nu$. In Fig. 3, **the vanilla OTM fails to learn the correct optimal transport plan in three cases except for the Parallel case**. OTM fails to cover the target measure $\nu$ in the Perpendicular and Multi-perpendicular cases. In the One-to-Many case, OTM does not learn the correct $T^\star$, i.e., the vertical transport.

On the other hand, as we can see from a comparison with Fig. 1, **our OTP successfully learns the optimal transport plan $\pi^\star$**. In particular, in the One-to-Many example, our model successfully recovers the correct stochastic transport map $\pi^\star(y|x)$ by utilizing the initial noise as guidance to either the upper or lower mode of the target distribution.

**Quantitative Comparison to Ground-truth** We evaluate the numerical accuracy of our OTP, SNOT with deterministic generator (OTM (Rout et al., 2022)), and SNOT with stochastic generator (OTM-s, Eq. 19), by comparing them to the closed-form ground-truth solutions. Here, we measure two metrics: the transport cost error $D_{cost} =$

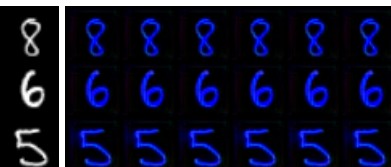

(a) OTM-s (FID=62.4, LPIPS=0.36)

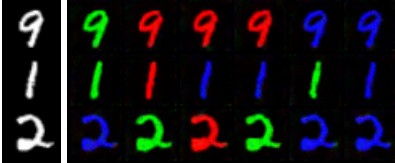

(b) Ours (FID=3.18, LPIPS=0.32)

*Figure 4.* **Experimental results on a stochastic transport map application**, i.e., MNIST-to-CMNIST translation.

*Table 2.* **Image-to-Image translation benchmark** results compared to existing Neural (Entropic) OT models. † indicates the results conducted by ourselves. DSBM scores are taken from (Gushchin et al., 2024; De Bortoli et al., 2024a).

| Data | Model | FID (↓) | LPIPS (↓) |
|---|---|---|---|
| Male-to-Female (64x64) | NOT (Korotin et al., 2023b) | 11.96 | - |
| | OTM† (Fan et al., 2022) | 6.42 | **0.16** |
| | DIOTM† (Choi et al., 2024a) | **4.48** | 0.20 |
| | OTP (Ours) | 4.75 | 0.20 |
| Wild-to-Cat (64x64) | DSBM (Shi et al., 2024) | 20+ | 0.59 |
| | OTM† (Fan et al., 2022) | 12.42 | 0.47 |
| | DIOTM† (Choi et al., 2024a) | 10.72 | **0.45** |
| | OTP (Ours) | **9.66** | 0.52 |
| Male-to-Female (128x128) | DSBM (Shi et al., 2024) | 37.8 | 0.25 |
| | ASBM (Gushchin et al., 2024) | 16.08 | - |
| | OTM† (Fan et al., 2022) | 7.55 | **0.21** |
| | DIOTM† (Choi et al., 2024a) | 7.40 | 0.25 |
| | OTP (Ours) | **6.38** | 0.27 |

$|W_2^2(\mu, \nu) - \int \|T_\theta(x) - x\|^2 d\mu(x)|$ and the target distribution error $D_{target} = W_2^2(T_{\theta\#}\mu, \nu)$. $D_{cost}$ assesses whether the model achieves the optimal transport cost, while $D_{target}$ measures how accurately the model generates the target distribution. Both models are tested on two synthetic datasets, Perpendicular and One-to-many (Fig. 3), with generalized dimensions of $d \in \{2, 4, 16, 64, 256\}$ (See Appendix E.1 for dataset details.).

Tab. 1 presents the quantitative results on the accuracy of the learned optimal transport plan $\pi_\theta$. Our OTP consistently achieves comparable or superior performance compared to both OTM and OTM-s across all metrics, particularly in high-dimensional settings. Note that these experimental results confirm the challenges of the existing SNOT framework in accurately recovering the target distributions, as discussed in Sec. 3.2.1 and 3.2.2. Specifically, as shown in $D_{target}$, OTM and OTM-s models exhibit significantly larger target distribution errors in higher dimensions.

### 5.2. Neural OT Evaluation on Unpaired Image-to-Image Translation Tasks

In this section, **we evaluate our model on the unpaired image-to-image translation task.** The image-to-image translation is one of the most widely used machine learning tasks in Neural OT models. The optimal transport map $T^\star$ (or plan $\pi^\star$) can be understood as a generator of target distributions, mapping an input $x$ to a *similar* counterpart $y$ by minimizing the transport cost $c(x, y)$. This mapping can be deterministic ($y = T^\star(x)$) or stochastic ($y \sim \pi^\star(\cdot|x)$). Therefore, the optimal transport map can naturally serve as a model for unpaired image-to-image translation.

**MNIST-to-CMNIST**  First, we demonstrate that our OTP model can learn stochastic transport mappings at the image scale. Specifically, we test our model on the MNIST-to-CMNIST translation task (See Appendix E.2 for dataset details). In this task, our Colored MNIST (CMNIST) dataset

consists of three colored variations (Red, Green, and Blue) for each grayscale image from the MNIST dataset (Fig. 2). Consequently, the desired OP plan should stochastically map each grayscale digit image to a colored digit image of the same digit type (Fig. 2).

Fig. 4 illustrates the experimental results. Here, we introduced a stochastic generator to OTM (OTM-s) to provide the capacity to learn a stochastic transport map. However, OTM exhibited the mode collapse problem, transporting all grayscale images to blue-colored images. On the other hand, our OTP successfully learns the optimal transport plan $\pi^\star$, achieving a stochastic mapping to Red, Green, and Blue colors. This phenomenon is also observed in the quantitative metrics. Our model significantly outperforms OTM in FID score (↓) (3.18 vs. 62.4) and archives a better score in LPIPS (↓) (0.32 vs. 0.36).

**Image-to-Image Translation**  We assess our model on three image-to-image translation benchmarks: *Male-to-Female (Liu et al., 2015)* ($64 \times 64$, $128 \times 128$) and *Wild-to-Cat (Choi et al., 2020)* ($64 \times 64$). For comparison, we include several OT models (*NOT, OTM, and DIOTM*) and Entropic OT models (*DSBM and ASBM*).

Tab. 2 presents the quantitative scores for the image-to-image translation tasks (See Appendix F.4 for qualitative examples). We adopted the FID (Heusel et al., 2017) and LPIPS (Zhang et al., 2018) scores for quantitative evaluation. Note that these FID and LPIPS scores serve similar roles as $D_{cost}$ and $D_{target}$ in Sec 5.1, respectively. Our primary evaluation metric is the FID score because it measures how well the translated images align with the target semantics. As shown in Tab. 2, our model demonstrates state-of-the-art FID scores and competitive LPIPS scores compared to existing (entropic) Neural OT models. Specifically, in the Male-to-Female ($128 \times 128$) task, our OTP model achieves a FID score of 6.38, outperforming the SNOT model (OTM), the other Neural OT model (DIOTM), and entropic OT

models (DSBM and ASBM). Although OTM achieves a lower but comparable LPIPS score (0.21), its significantly worse FID score (7.55) suggests large target semantic errors. Thus, we prioritize FID as our primary metric.

## 6. Conclusion

In this paper, we provided the first theoretical analysis of the sufficient condition that prevents the spurious solution issue in Semi-dual Neural OT. Based on this analysis, we proposed our OTP model for learning both the OT Map and OT Plan, even when this sufficient condition is not satisfied. Our experiments demonstrated that OTP successfully recovers the correct OT Plan when existing models fail and achieves state-of-the-art performance in unpaired image-to-image translation. Our primary contribution is improving Neural OT frameworks by addressing their fundamental limitations, i.e., failing to recover the correct OT Map even with the ideal max-min solution. One limitation of our work is that our convergence theorem holds up to a subsequence (Thm. 4.1). Nevertheless, in practice, our gradual training scheme (Alg 1) did not show any convergence issues. Also, our analysis provides a sufficient condition, rather than a necessary and sufficient one (Thm. 3.1), leaving room for further refinement in understanding the exact conditions under which spurious solutions occur.

## Acknowledgements

JW was partially supported by the National Research Foundation of Korea(NRF) grant funded by the Korea government(MSIT) [RS-2024-00349646]. DK was partially supported by the National Research Foundation of Korea (NRF) grant funded by the Korea government (MSIT) (No. RS-2023-00252516 and No. RS-2024-00408003), the POSCO Science Fellowship of POSCO TJ Park Foundation, and the Korea Institute for Advanced Study. JM is supported by National Research Foundation of Korea (NRF) grant funded by the Korea government (MSIT) [RS-2024-00410661].

## Impact Statement

This paper presents work whose goal is to advance the field of Machine Learning. There are many potential societal consequences of our work, none of which we feel must be specifically highlighted here.

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

# A. Proofs

## A.1. A generalized version of Theorem 3.1

In this section, we introduce theorems in Villani et al. (2009) and prove a generalized version of Thm. 3.1 (See Thm. A.6). We believe that rediscovering and summarizing this generalized version in the machine learning literature will help point out other possible directions for developing algorithms to avoid spurious solutions.

To begin with, we specify the assumptions on the domain:

**A0.** We consider $\mathcal{X} \subset M$, where $M$ is a smooth complete connected Riemannian manifold and $\mathcal{X}$ is a closed subset of $M$, with $\dim(\partial \mathcal{X}) \leq d - 1$. $\mathcal{Y}$ is an arbitrary Polish space.

The following two lemmas, Lemma A.1 and Lemma A.5, play a pivotal role in the proof of Thm. 3.1. Furthermore, we introduce a generalized version of the theorem.

We begin with Lemma A.1, which establishes the existence of a Kantorovich potential and characterizes its optimality.

**Lemma A.1** (Theorem 5.10 in (Villani et al., 2009)). *Let $(\mathcal{X}, \mu)$ and $(\mathcal{Y}, \nu)$ be two Polish probability spaces and let $c : \mathcal{X} \times \mathcal{Y} \to \mathbb{R}$ be a lower semi-continuous cost function such that is lower bounded. Suppose the optimal cost $C(\mu, \nu) := \inf_{\pi \in \Pi(\mu, \nu)} \int c d\pi$ is finite. Then,*

$$C(\mu, \nu) = \max_{V \in S_c} \left( \int_{\mathcal{X}} V^c(x) d\mu(x) + \int_{\mathcal{Y}} V(y) d\nu(y) \right). \tag{22}$$

*In other words, there exists a c-concave function $V$ that makes strong duality (Eq. 22) satisfy. Moreover, for c-cyclically monotone set $\Gamma \subset \mathcal{X} \times \mathcal{Y}$, for any $\pi \in \Pi(\mu, \nu)$ and c-concave function $\psi$,*

$$\pi \text{ is optimal } \Leftrightarrow \pi(\Gamma) = 1, \qquad \psi \text{ is optimal } \Leftrightarrow \Gamma \subset \partial_c \psi. \tag{23}$$

**Assumptions on the Cost Function** Throughout the discussion, we assume that the cost function $c(\cdot, \cdot)$ satisfies the following assumptions:

**A1.** $c : M \times \mathcal{Y} \to \mathbb{R}$ is a continuous cost function that is bounded below.

**A2.** $c$ is superdifferentiable everywhere.

**A3.** $\nabla_x c(x, \cdot)$ is injective where defined.

**A4.** $c$ satisfies either the **SC** condition (see Definition A.2) or the $\mathbf{H}_\infty$ condition (see Definition A.3).

Note that Assumption **A4** is required to ensure the existence of a nonempty $c$-subdifferential, i.e., $|\partial_c \psi| \geq 1$. We now provide the definitions of **SC** and $\mathbf{H}_\infty$.

**Definition A.2** (SC). We call the cost function $c : \mathcal{X} \times \mathcal{Y} \to \mathbb{R}$ is **SC** if $c(x, y)$ is locally semi-concave as a funtion of $x$, uniformly in $y$, i.e.

$$\exists M \in \mathbb{R} \quad \text{s.t.} \quad c(x, y) - M\|x\|^2 \text{ is concave for all } y \in \mathcal{Y}. \tag{24}$$

Here, **SC** stands for semi-concavity.

**Definition A.3** ($\mathbf{H}_\infty$). We say the cost function $c : \mathcal{X} \times \mathcal{Y} \to \mathbb{R}$ satisfies the condition $\mathbf{H}_\infty$ if the cost $c$ satisfies the following two conditions:

**H1.** For any $x$ and for any measurable set $S$ whose tangent space $T_x S$ is not contained in a half-space, there exists a finite collection of elements $z_1, \dots, z_k \in S$ and a small ball $B$ containing $x$, such that for any $y$ outside of a compact set,

$$\inf_{w \in B} c(w, y) \geq \inf_{1 \leq j \leq k} c(z_j, y).$$

**H2.** For any $x$ and any neighborhood $U$ of $x$, there exists a small ball $B$ containing $x$ such that

$$\lim_{y \to \infty} \sup_{w \in B} \inf_{z \in U} [c(z, y) - c(w, y)] = -\infty.$$

*Remark* A.4. Both **H1** and **H2** describe the behavior of the cost function $c(x, y)$ when $y \to \infty$. Consequently, these two conditions are automatically satisfied when $\mathcal{Y}$ or the support of $\mu$ is a compact set.

Next, we state and prove a generalized version of our main theorem, Theorem 3.1. We start with the following lemma:

**Lemma A.5** (Thm. 10.28 and Thm. 10.42 in (Villani et al., 2009)). *Assume A0-A4. Let $\mu \in P(\mathcal{X})$ and $\nu \in P(\mathcal{Y})$ such that*

**A5.** *The optimal cost $C(\mu, \nu) := \inf_{\pi \in \Pi(\mu,\nu)} \int_{\mathcal{X} \times \mathcal{Y}} c(x, y) d\pi(x, y)$ is finite;*

**A6.** *Any $c$-convex function is differentiable $\mu$-almost surely ($\mu$-a.s.) on its domain of $c$-subdifferentiability.*

*Then, there exists a unique deterministic optimal coupling $\pi \in \Pi(\mu, \nu)$ in law. Moreover, there exists a $c$-concave function $\psi$ such that*

$$\nabla_x c(x, y) - \nabla\psi(x) = 0, \quad \text{for } \pi\text{-almost surely.} \tag{25}$$

*In other words, the Monge map $T^\star$ exists, and satisfies $\nabla_x c(x, T^\star(x)) - \nabla\psi(x) = 0$, $\mu$-a.s.. Moreover, if $T : \mathcal{X} \to \mathcal{Y}$ satisfies*

$$T(x) \in \{y \in \mathcal{Y} : \nabla_x c(x, y) - \nabla\psi(x) = 0\}, \tag{26}$$

*then $T$ is the unique Monge map (in law).*

We now state and prove the generalized version of Thm. 3.1. Then, we discuss how this generalized theorem reduces to Thm. 3.1 under the standard quadratic cost setting.

**Theorem A.6.** *Assume A0-A6.*

(1) *Then, there exists a unique OT Map $T^\star$ in (Eq. 3) and a (possibly non-unique) Kantorovich potential $V^\star \in S_c$ in (Eq. 5).*

(2) *For the Kantorovich potential $V^\star \in S_c$, a solution of the minimization problem,*

$$T(x) \in \mathcal{D}_x := \arg\inf_{y \in \mathcal{Y}} \{c(x, y) - V^\star(y)\} \tag{27}$$

*is uniquely determined $\mu$-a.s., i.e. $\mathcal{D}_x = \{y_x\}$ for $\mu$-a.s $x \in \mathcal{X}$. In particular, a map $T : \mathcal{X} \to \mathcal{Y}$ given by $x \mapsto y_x \in \mathcal{D}_x$ is a unique OT Map $T^\star$ $\mu$-a.s..*

*Proof.* From Lemma A.1 and A.5, there exists a unique Monge map $T^\star$ (in law) and a Kantorovich potential $V^\star \in S_c$. Let $\psi := (V^\star)^c$. Because $((V^\star)^c, V^\star)$ is a pair of Kantorovich potentials, Lemma A.1 implies:

$$T^\star(x) \in \partial_c(V^\star)^c(x) := \{y \in \mathcal{Y} : (V^\star)^{cc}(y) = c(x, y) - (V^\star)^c(x)\}, \tag{28}$$

Note that $\psi := V^c$ is a $c$-concave, hence, by assumption **A6**, it is differentiable $\mu$-a.s.. For every $y \in \partial_c(V^\star)^c(x)$, differentiating both sides of the equation $(V^\star)^{cc}(y) = c(x, y) - (V^\star)^c(x)$ with respect to $x$ yields:

$$0 = \nabla_x c(x, y) - \nabla(V^\star)^c(x) = \nabla_x c(x, y) - \nabla\psi(x). \tag{29}$$

Therefore, the condition in Lemma A.5 (i.e., Eq. 25) is satisfied. From the last statement of Lemma A.5, the $c$-subdifferential $\partial_c\psi(x)$ is unique (i.e., $|\partial_c\psi(x)| = 1$) for all $x \in \mathcal{X}\backslash Z$, where $Z$ denotes the set of points where $\psi$ is non-differentiable. Because $\psi$ is differentiable $\mu$-almost surely, $Z$ has a zero measure with respect to $\mu$. Therefore, the mapping $x \mapsto \partial_c(V^\star)^c(x)$ is uniquely defined $\mu$-almost surely.

Now, consider:

$$\inf_{y \in \mathcal{Y}} (c(x, y) - V^\star(y)) = (V^\star)^c(x) = c(x, y) - (V^\star)^{cc}(y) \quad \text{for } y_0 \in \partial_c(V^\star)^c(x), \tag{30}$$

where the first equality follows from the definition of $c$-transform and the second equality comes from Eq. 28. Since $V^\star$ is assumed to be $c$-concave, we have $(V^\star)^{cc} = V^\star$. Therefore, the minimizer in

$$T(x) \in \arg\min_{y \in \mathcal{Y}} \{c(x, y) - V^\star(y)\} \tag{31}$$

is unique $\mu$-almost surely, due to the uniqueness of $\partial_c(V^\star)^c(x)$. By Lemma A.5, this uniquely defined $T$ is a Monge map. $\square$

## A.2. Proofs of Thm. 3.1 and Thm. 3.2

In this section, we prove Thm. 3.1 by verifying that all the assumptions **A0**–**A6** are satisfied. Moreover, Thm 3.1 is a specific realization of Theorem A.6.

*Proof of Thm. 3.1.* **Step 1. Check the basic assumptions:** Since $\mathcal{X} = \mathcal{Y}$ are closures of connected opensets, it trivially satisfies the domain assumption (**A0**). Assumption **A6** is the assumption of the theorem. Since $\mu$ and $\nu$ have finite second moments, the optimal transport cost is finite (**A5**). Furthermore, since $c$ is a quadratic cost, it is continuous, bounded below (**A1**), superdifferentiable (**A2**), and $\nabla_x c(x, \cdot)$ is injective for every $x \in \mathcal{X}$ (**A3**). Trivially, $c$ is semi-concave with respect to $x$ (**A4**).

**Step 2. Realization of Assumption A6:** Assumption **A6** can be satisfied in various ways. For completeness, we refer to the theoretical results outlined in Remark 10.33 of Villani et al. (2009), which provide the following sufficient conditions for Assumption **A6**:

**R1.** $c$ is Lipschitz on $\mathcal{X} \times \mathcal{Y}$ and $\mu$ is absolutely continuous.

**R2.** $c$ is locally Lipschitz, $\mu, \nu$ are compactly supported and $\mu$ is absolutely continuous.

**R3.** $c$ is locally semi-concave and satisfies the $\mathbf{H}_\infty$ condition, and $\mu$ does not give mass to to sets of Hausdorff dimension at most $d - 1$.

In our case, the cost function is quadratic, and therefore hence it is semi-concave. Moreover, as discussed in Step 2, the quadratic cost satisfies the $\mathbf{H}_\infty$ condition. Finally, our assumption that $\mu$ does not assign positive mass to sets of Hausdorff dimension at most $d - 1$ ensures that condition **R3** is satisfied. Thus, all assumptions required by Thm. A.6 are met under the conditions of Thm. 3.1. Therefore, the set $\mathcal{D}_x$ is a singleton $\mu$-almost surely.

$\square$

**Other Realization of the Cost Function**  Thm. A.6 is formulated in the general formulation with respect to the cost function. Thus, our theorem can be applied to various cost functions other than the standard quadratic cost. In this paragraph, we present several alternative realizations of cost functions that satisfy the required assumptions. We assume $\mathcal{X}, \mathcal{Y} \subset M = \mathbb{R}^d$.

**R4.** **A1**, **A2** and **A3** are satisfied when $c$ is $C^1$ strictly convex function.

**R5.** On top of **R4**, if $c$ also has bounded Hessian and $\mu$ does not assign mass to sets of Hausdorff dimension at most $d - 1$, then $c$ is **SC**, thus satisfies **A4**. Moreover, by Example 10.35 in Villani et al. (2009), this setting also satisfies **A6**.

**R6.** Suppose $c(x, y) = h(x - y)$ where $h : \mathbb{R}^d \to \mathbb{R}$ is a convex, superlinear, strictly increasing with respect to $|x - y|$, and radially symmetric function. Then, $c$ satisfies the $\mathbf{H}_\infty$ condition, hence satisfies **A4** (See Example 10.19-10.21 in Villani et al. (2009)).

**R7.** **R6** combined with **R3** and **R4**, all the assumptions are satisfied if: (i) $c(x, y) = h(x - y)$ is a $C^1$ strictly convex and locally semi-concave function where (ii) $h$ is a radially symmetric and strictly increasing function with respect to $|x - y|$, (iii) $\mu$ does not assign mass to sets of Hausdorff dimension at most $d - 1$.

**R8.** If the domains $\mathcal{X}$ and $\mathcal{Y}$ are both compact, then **A4** is automatically satisfied by the definition of $\mathbf{H}_\infty$. Thus, combining **R2** and **R4**, i.e. (i) $\mathcal{X}$ and $\mathcal{Y}$ are compact, (ii) $\mu$ is absolutely continuous, and (iii) $c$ is $C^1$ strictly convex function, then all the assumptions are satisfied.

In summary, the following proposition outlines various conditions that ensure all assumptions are met:

**Proposition A.7.** *Let $\mathcal{X}, \mathcal{Y} \subset M = \mathbb{R}^d$ satisfy Assumption **A0**. Suppose that any of the following conditions hold:*

1. *$c$ is $C^1$ strictly convex with bounded Hessian. Moreover, $\mu$ does not assign mass to sets of Hausdorff dimension at most $d - 1$;*

2. $\mathcal{X}$ and $\mathcal{Y}$ are compact, $\mu$ is absolutely continuous, and $c$ is $C^1$ a strictly convex function;

3. $c(x, y) = h(x - y)$ is a $C^1$ strictly convex and locally semi-concave function, where $h$ is a radially symmetric and strictly increasing function with respect to $\|x - y\|$, and $\mu$ does not charge sets of dimension $d - 1$.

*Then, all assumptions (A1–A6) in Theorem A.6 are satisfied.*

We emphasize that extending this analysis to Polish spaces $(\mathcal{X}, \mu)$ and $(\mathcal{Y}, \nu)$, with cost functions $c : \mathcal{X} \times \mathcal{Y} \to \mathbb{R}$ satisfying the above conditions, presents a promising direction for future research and offers significant potential for theoretical advancement.

**Proof of Theorem 3.2**   Thm. 3.2 is can be directly proved by leveraging Cor. 4 in Staudt et al. (2022):

*Proof of Thm. 3.2.* Let $K \subset \mathcal{X}$ be the compact set. Since $\mathcal{Y}$ is also compact, there exists $R > 0$ such that $K, \mathcal{Y} \subset B_R(0)$. Then, for

$$\|c(x, y_1) - c(x, y_2)\| \leq |x \cdot (y_1 - y_2)| + \frac{1}{2}|\|y_1\|^2 - \|y_2\|^2| \leq R\|y_1 - y_2\| + \frac{1}{2}2R\|y_1 - y_2\| = 2R\|y_1 - y_2\|. \quad (32)$$

Thus, by using Thm. 5 of Staudt et al. (2022), our assumptions satisfy the conditions of Corollary 4 in Staudt et al. (2022). Therefore, there exists a unique $c$-concave Kantorovich potential. $\square$

Note that Corollary 3.3 can be easily proved by simply combining Thm.s 3.1 and 3.2.

## B. Non-convergence of Stochastic Parametrization in Semi-dual Neural OT

In this section, we further elaborate Prop. 3.4. Let $\pi^\dagger(y|x)$ denote the conditional distribution induced by $T_\theta(x, \cdot)$ : $(Z, \mathcal{N}(0, I)) \to \mathcal{Y}$ where

$$T_\theta(x, z) \in \arg\min_{y \in \mathcal{Y}}\{c(x, y) - V^\star(y)\}, \quad (x, z) \sim \mu \times \mathcal{N}(0, I)\text{-a.s.}. \quad (33)$$

Since the subdifferential of the $c$-transform of $V^\star$ is defined as

$$\partial_c(V^\star)^c(x) := \{y \in \mathcal{Y} : V^c(x) = c(x, y) - V^\star(y)\} = \arg\min_{y \in \mathcal{Y}}\{c(x, y) - V^\star(y)\}, \quad (34)$$

Eq. 33 can be rewritten as follows:

$$\pi^\dagger(\partial_c(V^\star)^c(x) \mid x) = 1, \quad \mu\text{-a.s..} \quad \Leftrightarrow \quad \pi^\dagger \in P(\mathcal{X} \times \mathcal{Y}) \text{ satisfies } \pi^\dagger(\partial_c V^\star) = 1. \quad (35)$$

Here, $V^\star \in S_c$ represents the optimal potential and $\partial_c V^\star := \{(x, y) \in \mathcal{X} \times \mathcal{Y} : (V^\star)^c(x) + V(y) = c(x, y)\}$. In summary, the stochastic map optimization problem in Eq. 33 is equivalent to finding the joint distribution $\pi^\dagger \in P(\mathcal{X} \times \mathcal{Y})$ that satisfies $\pi^\dagger(\partial_c V^\star) = 1$.

However, in general, this condition $\pi^\dagger(\partial_c V^\star) = 1$ does not guarantee that $\pi$ is the optimal transport plan for the Kantorovich's problem (Eq. 4). As discussed in Lemma A.1, under the mild assumptions on the cost function $c$, $\pi^\dagger$ is optimal if and only if $\pi^\dagger(\Gamma) = 1$ for $c$-cyclic monotone $\Gamma$. Since $\Gamma \subset \partial_c V^\star$, we can say that one of the solution $\pi^\dagger$ is the optimal transport plan, however, the converse may not satisfy (See failure cases in Sec. 3.2.2). Additionally, if $\partial_c(V^\star)^c(x)$ is unique $\mu$-almost surely, then there is a unique (in law) deterministic optimal coupling of $(\mu, \nu)$ (See Thm. 5.30 in Villani et al. (2009)). The discussion above can be summarized as follows:

**Proposition B.1** (Formal). *Suppose the assumptions of Lemma A.1 hold. Let $V^\star \in S_c$ be the Kantorovich potential. For $\pi^\dagger \in P(\mathcal{X} \times \mathcal{Y})$, the condition $\pi^\dagger(\partial_c V^\star) = 1$ does not imply that $\pi^\dagger$ is an optimal transport plan.*

## C. Convergence of OTP

We present the convergence theorem for the optimal transport plan from Villani et al. (2009). Thm. 4.1 is a direct consequence of Thm. 3.1 and Thm. C.1.

**Theorem C.1** ([Villani et al. (2009)](), Thm. 5.20). *Let $c \geq 0$ be a real-valued, continuous, and lower-bounded cost function. Consider a sequence of continuous cost functions $\{c_k\}_{k\in\mathbb{N}}$ that uniformly converges to $c$. Let $\{\mu_k\}_{k\in\mathbb{N}}$ and $\{\nu_k\}_{k\in\mathbb{N}}$ be sequences of probability measures that weakly converge to $\mu$ and $\nu$, respectively. For each $k$, let $\pi_k^\star$ be an optimal transport plan between $\mu_k$ and $\nu_k$. If $\int c_k d\pi_k^\star < \infty$, then, up to the extraction of a subsequence, $\pi_k^\star$ converges weakly to some $c$-cyclically monotone transport plan $\pi^\star \in \Pi(\mu, \nu)$. Moreover, if*

$$\liminf_{k\in\mathbb{N}} \int c_k d\pi_k^\star < \infty, \tag{36}$$

*then the optimal transport cost between $\mu$ and $\nu$ is finite and $\pi^\star$ is an optimal transport plan.*

**Corollary C.2** (Corollary 4.1). *Consider a sequence absolutely continuous probability measures $\{\mu_{\epsilon_k}\}_{k\in\mathbb{N}}$ such that $\{\mu_{\epsilon_k}\}$ weakly converges to $\mu$. Then, up to the extraction of a subsequence, our OTP model, utilizing $\{\mu_{\epsilon_k}\}_{k\in\mathbb{N}}$, weakly converges to the optimal transport plan $\pi^\star$ between $\mu$ and $\nu$.*

*Proof of Corollary 4.1.* The absolutely continuous measure $\mu_{\epsilon_k}$ does not assign positive mass to measurable sets of Hausdorff dimension at most $d - 1$. Therefore, by Thm. 3.1, our OTP model for noise level $\epsilon_k$ (Eq. 20) correctly recovers the optimal transport plan $\pi_k^\star = (Id, T_k^\star)_{\#}\mu_{\epsilon_k}$. Then, by Thm. C.1, $\pi_k^\star$ weakly converges to the optimal transport plan $\pi^\star$ between $\mu$ and $\nu$. $\square$

# D. Related Works

## D.1. Spurious Solution Problem

In this section, we overview previous attempts to address the spurious solution problem in the Semi-dual Neural OT approaches. The spurious solution problem refers to the problem where the solution of the max-min learning objective $\mathcal{L}_{V_\phi, T_\theta}$ of Semi-dual Neural OT fails to fully characterize the correct optimal transport map. [Rout et al. (2022)](); [Fan et al. (2022)]() first identified this issue. They proved that while true optimal potential and transport map $(V^\star, T^\star)$ are the solution to the max-min problem $\mathcal{L}_{V_\phi, T_\theta}$ (Eq. 9), the reverse does not hold. Furthermore, [Fan et al. (2023)]() proved that, when $\mu$ is atomless, the saddle point solution $(V_{sad}^\dagger, T_{sad}^\dagger)$ (Eq. 37) of $\mathcal{L}_{V_\phi, T_\theta}$ recovers the optimal transport map:

$$V_{sad}^\dagger \in \arg\max_V \mathcal{L}(V, T_{sad}^\dagger), \ T_{sad}^\dagger \in \arg\min_T \mathcal{L}(V_{sad}^\dagger, T). \tag{37}$$

However, this does not hold for the max-min solution of $\mathcal{L}_{V_\phi, T_\theta}$. In this paper, we establish the sufficient condition under which the max-min solution recovers the optimal transport map (Thm. 3.1). Furthermore, we suggest a method for learning the optimal transport plan $\pi^\star$, which is applicable even when the optimal transport map $T^\star$ does not exist (Sec 4).

For the weak OT problem, [Korotin et al. (2023b;a)]() provided a theoretical analysis for spurious solutions in semi-dual approaches. Note that *these analyses were conducted for the $\gamma$-weak quadratic cost with $\gamma > 0$ and, therefore, do not cover the standard OT problem.* Additionally, [Korotin et al. (2023a)]() proposed an alternative approach that modifies the cost function by introducing a positive definite symmetric kernel. While this approach addresses the spurious solution issue in the weak OT problem, it solves an inherently different problem due to the modified cost function. In contrast, our work is **the first attempt to analyze the conditions under which spurious solutions occur in the standard OT problem** (Eq. 4).

## D.2. Neural Optimal Transport Models

For completeness, we provide an overview of various approaches for Neural OT solvers. First of all, several works proposed methods based on the semi-dual formulation of the OT problem ([Rout et al., 2022](); [Fan et al., 2022](); [Choi et al., 2023](); [Makkuva et al., 2020](); [Fan et al., 2023]()). The spurious solution problem occurring in these formulations is the main focus of this paper.

Moreover, there are dynamic Neural OT models, such as bridge matching ([Shi et al., 2024](); [De Bortoli et al., 2024b](); [Liu et al., 2022]()) and flow matching ([Tong et al., 2024a;b](); [Lipman et al., 2023]()). These models learn continuous dynamics that transport the source distribution to the target distribution. Particularly, the bridge matching methods learn stochastic transport between two distributions based on the Entropic Optimal Transport (EOT) problem, i.e., the OT problem with an entropic regularizer. In contrast, our OTP model learns stochastic OT plans derived from the standard (non-entropic) OT problem, making the target formulation fundamentally different.

Lastly, there are recent Neural OT models based on non-minimax objective functions (Uscidda & Cuturi, 2023; Kassraie et al., 2024). Uscidda & Cuturi (2023) introduces a regularizer known as the Monge gap to learn the OT map. Kassraie et al. (2024) proposes a gradual learning scheme by solving a sequence of EOT problems with decreasing entropic regularization. Although this gradual approach is conceptually related to our approach, Kassraie et al. (2024) solves the discrete OT problem, i.e., the OT problem between empirical distributions over finite samples. In contrast, our OTP model is designed for the continuous OT problem between two continuous distributions, where training data consists of i.i.d. samples drawn from these distributions.

## E. Implementation Details

### E.1. Synthetic Data Experiments

In this section, we explain the implementation details for the synthetic data experiments (Sec. 5.1), including dataset description, model architecture, and training hyperparameters.

**Dataset Description**    Throughout this paragraph, let $x, y \in \mathbb{R}^d$, and $n = d/2$. Then, let $x = (x_1, x_2)$ and $y = (y_1, y_2)$, where $x_1, x_2, y_1, y_2 \in \mathbb{R}^n$. Moreover, let $e_1 = (1, 0, \ldots, 0) \in \mathbb{R}^n$.

- **Perpendicular**: We generate $x \sim \mu$ as follows: $x_1 \sim U([-1, 1]^n)$, and $x_2 \equiv 0$. Similarly, we sample $y \sim \nu$ by $y_1 \equiv 0$ and $y_2 \sim U([-1, 1]^n)$.

- **Horizontal**: We generate $x \sim \mu$ as follows: $x_1 \sim U([-1, 1]^n)$, and $x_2 \equiv 0$. Similarly, we sample $y \sim \nu$ by $y_1 \sim U([-1, 1]^n)$ and $y_2 = e_1$.

- **One-to-Many**: We generate $x \sim \mu$ as follows: $x_1 \sim U([-1, 1]^n)$, and $x_2 \equiv 0$. Similarly, we sample $y \sim \nu$ by $y_1 \sim U([-1, 1]^n)$ and $y_2 \sim \text{Cat}((e_1, -e_1), (0.5, 0.5))$.

- **Multi-Perpendicular**: Let $\mathbb{P} := \text{Cat}\left((\frac{-3}{4}, \frac{-1}{4}, \frac{1}{4}, \frac{3}{4})e_1, (\frac{1}{4}, \frac{1}{4}, \frac{1}{4}, \frac{1}{4})\right)$ We generate $x \sim \mu$ by sampling $x_1 \sim U([-1, 1]^n)$, and $x_2 \sim \mathbb{P}$. Similarly, we sample $y \sim \nu$ by $y_1 \sim \mathbb{P}$ and $y_2 \sim U([-1, 1]^n)$.

**Training Details**    For every experiments, we share the same network architecture and the same training hyperparameters. We employ one-hidden layer with ReLU activations for both potential function $v_\phi$ and transport map $T_\theta$ parametrization. For experiments of $d = 2$ and $d = 4$, we use hidden dimension of 256. For higher dimensions, we use hidden dimension of 1024. We employ the batch size of 128, the number of iterations of 20K, the learning rate of $10^{-4}$, and Adam optimizer for $(\beta_1, \beta_2) = (0, 0.9)$. To improve the optimization of the inner loop in the dual formulation of equation (9), we perform 20 updates to $T_\theta$ for every single update to $v_\phi$, i.e. $K_T = 20$. For the experiment on $d = 256$, we use $\alpha = 0.01$ and $\lambda = 1$. Otherwise we set $\alpha = 1$ and $\lambda = 0$.

For our experiment, we generate the perturbed data $\hat{x}$ as follows: $\hat{x} = x + \sigma z$ where $x \sim \mu, z \sim \mathcal{N}(\mathbf{0}, \mathbf{I})$. Here, we schedule the noise level $\sigma$ from $\sigma_{max} = 0.2$ to $\sigma_{min} = 0.05$. We update noise every $2K$ iterations, by the linear interpolation between initial noise to terminal noise. Specifically, in the $k$-th iteration, the noise level $\sigma_k$ is

$$\sigma_k = (1 - t)\sigma_{max} + t\sigma_{min}, \quad t = (P \times [k/P] + 1)/K, \tag{38}$$

where $P = 2000$, $[\cdot]$ is the least integer function and $K$ is the total iteration number.

In the NOT (Korotin et al., 2023b) implementation, we introduce additional noise $\xi \sim \mathcal{N}(\mathbf{0}, \mathbf{I})$ into the network $T_\theta$. we set the dimensionality of the noise $\xi$ to match the input data dimension. We simply concatenate the noise and the data. This augmented input is then passed through the transport network $T_\theta$.

### E.2. Image Translation

**MNIST→CMNIST**    In this paragraph, we describe the implementation details of Fig. 4. We create red, green, and blue-colored MNIST datasets by isolating individual color channels. To achieve this, we assign the grayscale MNIST digit images to a single color channel (red, green, or blue) while setting the other two channels to zero. We use the image size of 32, the number of iterations of 50K, the batch size of 64, the learning rate of $10^{-4}$, Adam optimizer with $(\beta_1, \beta_2) = (0, 0.9)$, $\lambda = 10$, $K_T = 10$, $\alpha = 0.1$, $\sigma_{max} = 2$, $\sigma_{min} = 0.5$, and $P = 100$. Note that we don't use any other techniques such as learning rate scheduling, exponential moving avarage, dropout, and clip.

We adopt the network architectures of DCGAN (Radford et al., 2015), depicted as follows: For generator, we use UNet architecture. For the input embedding module, we pass the input through the convolution layer, batch norm layer and activation layer. We employ four of downsample modules and four upsample modules, For every downsample module, we pass through convolution, activation, average plloing layer. For the upsample modules, we pass the inputs through upsample module, convolution, batch norm, activation layers. Note that as original UNet, we use the skip connections. For the last module, we pass through one convolution layer. For the activation function, we employ leaky ReLU with slope of 0.2. Every convolutional layers are $3 \times 3$ convolutional layers. For average pooling, we use convolutional layer of $3 \times 3$ with stride of 2. For upsampling module, we simply use blockwise upsampling module.

For the potential network, we use three downsample module. The settings of downsample module is same as the downsample module of the generator. After the downsample modules, we flatten it and pass it through linear layer. For every modules, note that the channel number (or the feature number) is fixed to 256.

**Image-to-Image Translation**    In this paragraph, we describe the implementation details of Tab. 2. Most of the hyperparameter except the network architecture and the number of iterations are shared. For every experiments, we use the batch size of 64, the learning rate of $10^{-4}$, Adam optimizer with $(\beta_1, \beta_2) = (0, 0.9)$, $\lambda = 10$, $K_T = 1$, $\alpha = 0.001$, $\sigma_{max} = 2$, $\sigma_{min} = 0.2$, and $P = 100$. Note that we don't use any other techniques such as learning rate scheduling, exponential moving avarage, dropout, and clip. In the experiments for image size of 64, we follow the network architecture of the CIFAR-10 experiment in (Choi et al., 2023; 2024c). In the experiments for image size of 128, we follow the network architecture of the CelebA-HQ experiment in (Choi et al., 2023). For Wild→Cat ($64 \times 64$), Male→Female ($64 \times 64$), and Male→Female ($128 \times 128$), we take the number of iterations of 60K, 300K, and 500K, respectively.

Here, we use Variance-Preserving noise scheduling, which is illustrated as follows: For $x \sim \mu, z \sim \mathcal{N}(\mathbf{0}, \mathbf{I})$, we generate the perturbed data $\hat{x}$ at $k$-th iterations as follows: $\hat{x} = \sqrt{1 - \epsilon_k}x + \sqrt{\epsilon_k}z$ where $\epsilon_k$ is defined as follows:

$$\epsilon_k = 1 - \exp\left(-\frac{\sigma_{max} - \sigma_{min}}{2}t^2 - \sigma_{min}t\right), \quad t = 1 - (P \times [k/P] + 1)/K, \tag{39}$$

where $P = 100$, $[\cdot]$ is the least integer function and $K$ is the total iteration number.

**Evaluation metric**    We follow the evaluation metric of Choi et al. (2024a). Specifically, for the Male→Female translation task, we transform Male in the test dataset, and use the transformed samples and the test samples of the Female data to calculate the evaluation metrics. For Wild→Cat, we generate 5000 samples from the test data of Wild dataset, and use the transformed samples and the training samples of the Cat data to calculate the evaluation metrics.

# F. Additional Results

## F.1. Input Convex Neural Networks

As discussed in Eq. 9, the search space for the potential $V$ is constrained to $S_c$, the set of $c$-concave potential functions. Strictly speaking, the potential $V_\phi$ must be specifically parameterized to ensure it lies within $S_c$. To address this, we parameterize $V_\phi$ using an input convex neural network (ICNN) (Amos et al., 2017; Korotin et al., 2021a). The ICNN architecture $f_\phi : \mathcal{Y} \to \mathbb{R}$ is specifically designed with structural and weight constraints to enforce input convexity, satisfying the property: $tf_\phi(x_1) + (1-t)f_\phi(x_2) \geq f((1-t)x_1 + tx_2)$ for all $t \in [0, 1]$. Given that $c(x, y) = \alpha\|x - y\|^2$, $\alpha\|y\|^2 - V_\phi(y)$ should satisfy input convexity. To ensure this, we parametrize $V_\phi$ as follows:

$$V_\phi(y) = \alpha\|y\|^2 - f_\phi(y), \tag{40}$$

where $f_\phi$ is an ICNN. Using this parameterization, we conduct experiments on two-dimensional data $d = 2$ across several toy datasets. The results are presented in Fig. 5.

## F.2. Ablation Studies: Constant Noise Scheduling

In practice, our OTP model decreases the noise level until it reaches a small constant $\epsilon_{min} > 0$ (see Algorithm paragraph in Sec 4). Since the noise level is not reduced exactly to zero, an alternative approach is to train our OTP model directly at the minimum noise level $\epsilon_{min}$ instead of gradually decreasing it. To investigate this alternative, we conduct an ablation study on noise scheduling, specifically Constant Noise Scheduling (CNS).

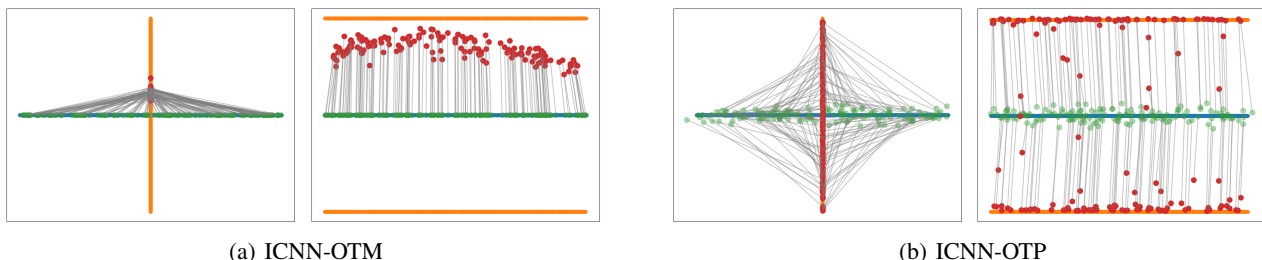

(a) ICNN-OTM                                    (b) ICNN-OTP

*Figure 5.* Visualization of failure cases with ICNN potential function $f_\phi(y) := \alpha\|y\|^2 - V_\phi(y)$. We compare the Optimal Transport map (a) and our OTP model (b) in the failure cases. The source data $x \sim \mu$, target data $y \sim \nu$, and generated data $T(x)$ are represented in Blue, Orange, and Red. As illustrated in the figure, OTM fails to transport source to the target, i.e. $T_{\#}\mu \neq \nu$. On the other hand, our model successfully transports source $\mu$ to target $\nu$.

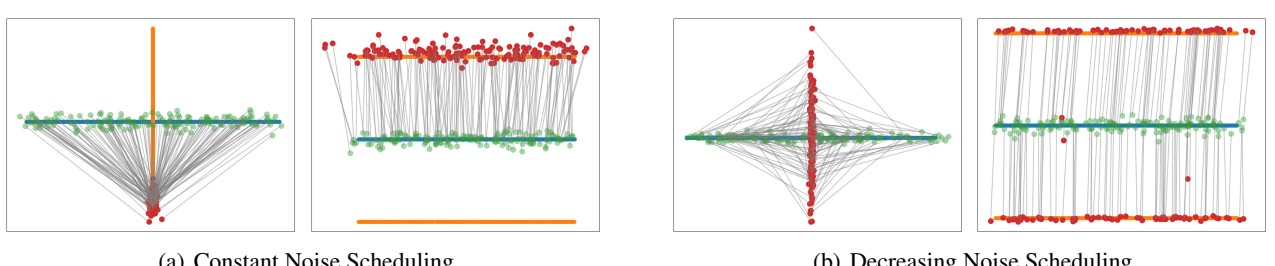

(a) Constant Noise Scheduling                   (b) Decreasing Noise Scheduling

*Figure 6.* We visualize the qualitative result between (a) constant noise scheduling, ($\sigma_{max} = \sigma_{min} = 0.05$) and (b) gradually decreasing noise scheduling ($\sigma_{max} = 0.2, \sigma_{min} = 0.05$). We illustrate the results on the data dimension $d = 64$. We select the 1st and 33rd axis to visualize the results effectively. The source data $x \sim \mu$, target data $y \sim \nu$, and generated data $T(x)$ are represented in Blue, Orange, and Red. The max-min solution fails to recover the correct OT map.

Fig 6 and Tab 3 present the results. In Fig 6, our OTP model with CNS scheduling (OTP-CNS) shows the mode collapse problem. In the left figure of Fig 6(a), OTP-CNS generates only the bottom part of the target distribution. Similarly, in the right figure of Fig 6(a), OTP-CNS covers only the upper part of the target distribution. In contrast, Fig 6(b) demonstrates that our original OTP model with decreasing noise scheduling successfully covers the entire target distribution without exhibiting the mode collapse problem. Tab 3 also shows this trend. Because of the mode collapse problem observed in Fig 6(a), OTP-CNS shows a significantly larger target distribution error $D_{target}$. This ablation study shows that our noise-decreasing scheme is a more effective choice for the OTP model.

### F.3. Benchmark Experiments

In this section, we present the experimental results on the Wasserstein-2 benchmark proposed in Korotin et al. (2021b). Specifically, we implemented our OTP model, based on the [MMv2:R] baseline in Korotin et al. (2021b), which solves the SNOT formulation (Eq. 20) using an ICNN architecture.

Note that this benchmark employs a Gaussian mixture as the source distribution, which is already absolutely continuous. Therefore, by our Thm 3.1, existing SNOT models can also recover the correct OT map. In this respect, this benchmark is slightly unfavorable to our OTP model, because our additional input noise would be regarded as an error. Nevertheless, as shown in Table 4, our model shows comparable performance to [MMv2:R].

*Table 3.* **Quantitative comparison of numerical accuracy** on synthetic datasets. The Const column stands for the constant noise scheduling, and Ours stands for our method which gradually decrease the noise level. Each model is evaluated by target distribution error $D_{target}(\downarrow)$

| Dimension | Model | Perpendicular | One-to-Many |
|---|---|---|---|
| $d = 16$ | Constant | 0.64 | 72.17 |
| | Ours | **0.59** | **0.65** |
| $d = 64$ | Constant | 12.36 | 20.72 |
| | Ours | **10.09** | **9.98** |

*Table 4.* **Quantitative comparison of numerical accuracy** on benchmark dataset proposed in (Korotin et al., 2021b).

| Dimension | D=16 | | D=64 | | D=256 | |
|---|---|---|---|---|---|---|
| Metric | $\mathcal{L}^2$-UVP ($\downarrow$) | cos ($\uparrow$) | $\mathcal{L}^2$-UVP ($\downarrow$) | cos ($\uparrow$) | $\mathcal{L}^2$-UVP ($\downarrow$) | cos ($\uparrow$) |
| L | 41.6 | 0.73 | 63.9 | 0.75 | 67.4 | 0.77 |
| QC | 47.2 | 0.70 | 75.2 | 0.70 | 88.2 | 0.66 |
| MM | 2.2 | 0.99 | 3.2 | 0.99 | 4.1 | 0.99 |
| MMv1 | 1.4 | 0.99 | 8.1 | 0.97 | 2.6 | 0.99 |
| MMv2 | 3.1 | 0.99 | 10.1 | 0.96 | 2.7 | 0.99 |
| MM-B | 6.4 | 0.96 | 13.9 | 0.94 | 22.5 | 0.93 |
| MMv2:R | 7.7 | 0.96 | 6.8 | 0.97 | 2.8 | 0.99 |
| OTP[†] | 8.7 | 0.97 | 6.5 | 0.97 | 3.7 | 0.99 |

## F.4. Additional Qualitative Results

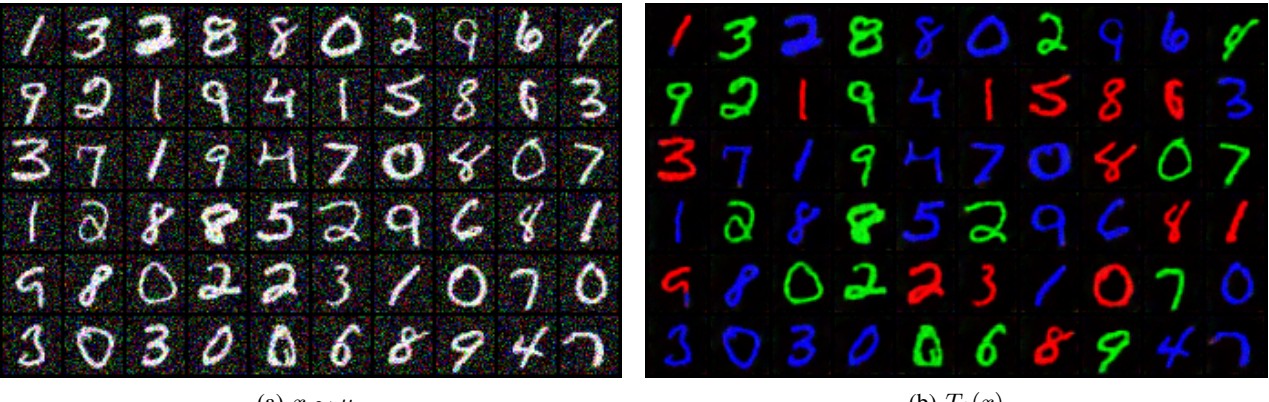

(a) $x \sim \mu_\epsilon$

(b) $T_\theta(x)$

*Figure 7.* Unpaired MNIST $\rightarrow$ CMNIST translation for $32 \times 32$ image.

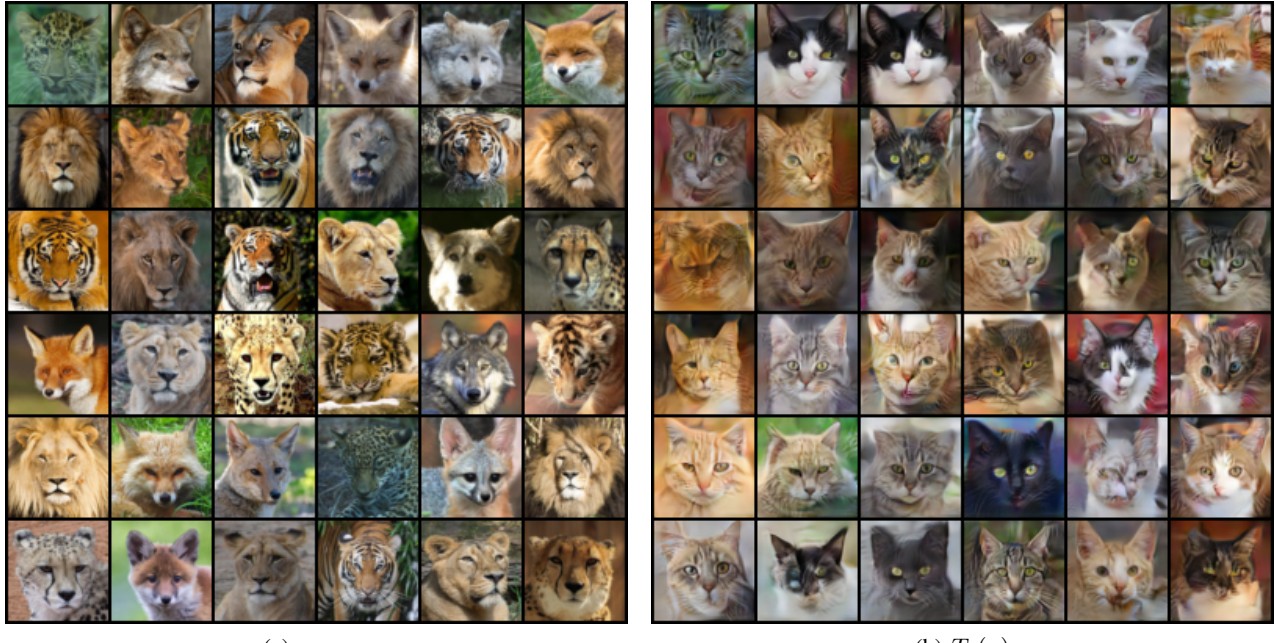

(a) $x \sim \mu_\epsilon$

(b) $T_\theta(x)$

*Figure 8.* Unpaired Wild $\rightarrow$ Cat translation for $64 \times 64$ image.

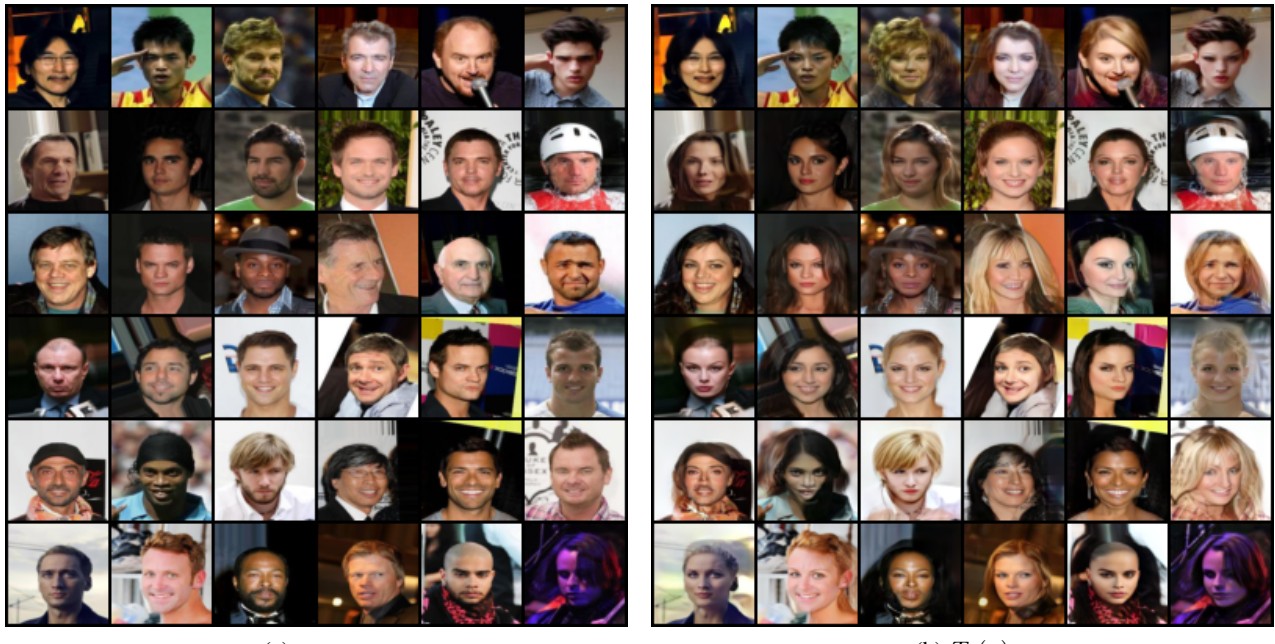

(a) $x \sim \mu_\epsilon$

(b) $T_\theta(x)$

*Figure 9.* Unpaired Male → Female translation for 64 × 64 image.

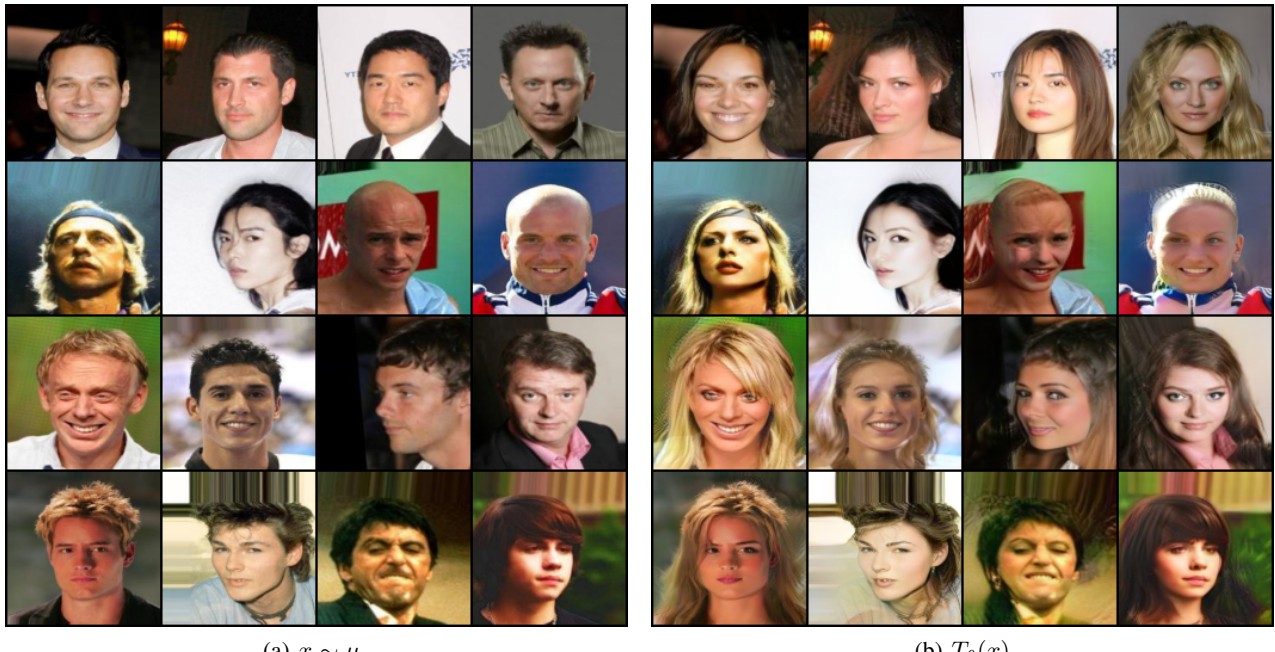

(a) $x \sim \mu_\epsilon$

(b) $T_\theta(x)$

*Figure 10.* Unpaired Male → Female translation for 128 × 128 image.

