# OpenReview forum: "Overcoming Spurious Solutions in Semi-Dual Neural Optimal Transport: A Smoothing Approach for Learning the Optimal Transport Plan"
_ICML.cc/2025/Conference — ICML 2025 poster_

### Official Review · Reviewer_v2np · 2025-03-12

**Overall Recommendation:** 4

**Summary:**

Semidual Neural OT (SNOT) is one of the NN-based formulation for computing transport maps for general OT problems. The paper seeks to address the issue of spurious solutions for SNOT formulation, a problem that has been known before. For example, when the source distribution places mass on low-dimensional sets, or when stochastic transport is needed (e.g., one-to-many mappings like colorization).. The paper also proposes a smoothing algorithm  to first smear the source distribution in order to satisfy the sufficient conditions they identified. Proofs and illustrative examples, and image-to-image translation examples are used to illustrate the effectiveness of the proposed method.

**Claims And Evidence:**

SNOT has been in existence for a while, but the paper is the first to identify a reasonable sufficient condition for the avoidance of spurious solutions for the minimax formulation.  The paper identifies that spurious solutions are avoided if the source distribution does not concentrate on sets with Hausdorff dimension <= d-1. This result is interesting but not surprising. The proposed smoothing algorithm is quite natural and the paper has done a good job illustrating its effectiveness. The paper also proves that the smoothed distributions weakly converge to the original, ensuring the learned transport plan approaches the true OT plan.

**Essential References Not Discussed:**

NA

**Experimental Designs Or Analyses:**

One interesting fact explained in the paper is that, the noise scheduling (gradual reduction) is critical; constant noise leads to mode collapse.

**Methods And Evaluation Criteria:**

NA

**Other Comments Or Suggestions:**

change "fake solution" to "spurious solution?"

**Other Strengths And Weaknesses:**

NA

**Questions For Authors:**

NA

**Relation To Broader Scientific Literature:**

NA

**Theoretical Claims:**

The identified sufficient condition is use for guiding the proposal of the smoothing algorithms.

---

> ### Author Rebuttal · Authors · 2025-03-29
>
> We sincerely thank the reviewer for carefully reading our manuscript and providing valuable feedback. We are especially grateful for the reviewer's recognition that our work is the **first to identify a reasonable sufficient condition for the avoidance of spurious solutions** for the minimax formulation used in SNOT. In addition, we greatly appreciate the reviewer' acknowledgment that our work has done **a good job illustrating the effectiveness** of the proposed smoothing algorithm.
>
> $ $
>
> ---
> ### **Experimental Designs Or Analyses:**
>
> $ $
>
> > **Q.** One interesting fact explained in the paper is that, the noise scheduling (gradual reduction) is critical; constant noise leads to mode collapse.
>
> **A.** Thank you for highlighting this point. We also found this phenomenon to be quite interesting. Our interpretation is that adding a certain level of noise to the input serves as a form of regularization, which helps stabilize the training of the OT map. Since our goal is to learn the OT Map that transports each source sample $x \sim \mu$ to a nearby target sample $y \sim \nu$, **the initial noise can guide the model to explore all modes of the target distribution**. Gradually reducing the noise then allows the model to refine the transport map without experiencing mode collapse.
>
> $ $
>
> ---
> ### **Other Comments Or Suggestions:**
>
> $ $
>
> > **Q.** Change "fake solution" to "spurious solution?"
>
> **A.** Thank you for suggesting the appropriate terms. We agree that "spurious solution" is a more appropriate term than "fake solution." We have **revised the title and updated the terminology** throughout the manuscript accordingly.

---

### Official Review · Reviewer_C1Wd · 2025-03-21

**Overall Recommendation:** 3

**Summary:**

The manuscript presents a minimax algorithm used to find solutions to the semi-dual optimal transport plan problem. A primary drawback of this method is that not all solutions of the semi-dual formulation yield optimal transport maps and optimal potential functions; in other words, the formulation can lead to “fake” solutions. The authors identify a sufficient condition on the source distribution to avoid such fake solutions. Based on this condition, they propose a new algorithm with theoretical guarantees for approximating the optimal transport plan.

**Claims And Evidence:**

The claims are supported by theoretical results, experiments, and references to relevant literature.

**Essential References Not Discussed:**

I cannot think of any essential references that are missing.

**Experimental Designs Or Analyses:**

The experiments and analyses are reasonable, including various comparisons and datasets.

**Methods And Evaluation Criteria:**

The method is theoretically founded, and the evaluation criteria include various metrics across different datasets. It would be beneficial to report standard deviations where possible and additional analysis on training time or complexity.

**Other Comments Or Suggestions:**

- Line 11. It has recently also been used with generative model like flow matching.
- Line 67, You could mention that T_\# is a push-forward operator.
- The citation formatting on line 100 seems odd.
- Potentially a missing word on line 141 "if OT Map T..."
- The phrasing of Thm 3.1 could be reviewed to improve clarity.
- Line 311 missing word "*be a sequence absolutely*"
- Overall the manuscript would benefit from another read and revision.
- I believe it is generally preferred to avoid starting a sentence with a mathematic symbol.
- It would be best to uniformize the table presentation, for example, in tab1 the number of decimals are not always the same.

**Other Strengths And Weaknesses:**

- The manuscript is overall well written, and the structure is very clear. I appreciated reading the examples, as they provide great insight while remaining straightforward.
- The experiments are interesting. While the datasets seem relatively simple, they include various settings and dimensions. Overall, the method appears to improve on existing approaches.
- The novelty of Theorem 3.1 is somewhat limited, as it follows from Lemma A.3. The proof verifies that the conditions of Lemma A.3 are satisfied; perhaps it should be regarded as a corollary instead.
- It would be beneficial to include a comparison of training complexity and time.

**Questions For Authors:**

- Could you comment on the fact that the uniqueness in Thm 3.1. is in law, is this a limitation ? Especially, is it the same definition of uniqueness that is used in example 1, 2 ?

- For the second row of figure one, with what algorithm do you compute the max-min solution ? Are the results stochastic ?

**Relation To Broader Scientific Literature:**

This work proposes a new method for learning an optimal transport (OT) map based on the semi-dual formulation. OT is utilized in various fields, including generative models and biology; thus, improving the quality of an estimator can have a significant impact.

**Theoretical Claims:**

I verified the proofs of Theorems 3.1 and 3.2 but did not verify the references cited in those proofs.

---

> ### Author Rebuttal · Authors · 2025-03-31
>
> We sincerely thank the reviewer for carefully reading our manuscript and providing valuable feedback. We greatly appreciate the reviewer's acknowledgment that our method is **theoretically founded**. Moreover, we’re pleased to hear that the reviewer **appreciated reading the examples, as they provide great insight while remaining straightforward.**
>
> We hope our responses to be helpful in addressing the reviewer's concerns.
>
> $ $
>
> ---
> ### **Other Strengths And Weaknesses:**
>
> $ $
>
> > **Q.** It would be beneficial to include a comparison of training complexity and time.
>
> **A.** We appreciate the reviewer for the insightful comment. **Due to the use of the gradual training scheme with noise scheduling, our OTP model requires more training iterations** compared to existing SNOT models. For example, as described in Appendix F, in our Male-to-Female(64x64) experiment, our model uses 300K training iterations, while the OTM uses 60K iterations. Since both models adopt the same backbone network, the training time ratio is the same as the training iteration ratio.
>
> However, we would like to emphasize that **OTM achieves its best performance when trained for 60K iterations**. In practice, the OTM model is known to suffer from training instability under long training [1]. Hence, in our experiments, increasing the number of iterations beyond 60K led to training failure with loss explosion, as observed in [1].
>
> Reference
> - [1] Choi, J., Choi, J., & Kang, M. Analyzing and Improving Optimal-Transport-based Adversarial Networks. ICLR 2024.
>
> $  $
>
> ---
> > **Q.** The novelty of Theorem 3.1 is somewhat limited, as it follows from Lemma A.3. The proof verifies that the conditions of Lemma A.3 are satisfied; perhaps it should be regarded as a corollary instead.
>
> **A.** Thank you for the thoughtful comment. We would like to clarify that Thm A.4 is an auxiliary result for proving Thm 3.1. Hence, Thm A.4 also contributes to the novelty of Thm 3.1. We chose to present Theorem 3.1 as a theorem to emphasize its **importance in the context of the Neural Optimal Transport**, where the quadratic cost function is widely adopted and the transport map is parametrized as follows (Eq 7 or 23):$$T_{\theta}: x \mapsto \arg \min_{y \in \cal{Y}} [c(x,y) - V_{\Phi}(y)]$$Thm A.4 shows that the **conclusion of Lemma A.3 (Eq. 22) implies that this transport map parametrization in SNOT (Eq 23) recovers the correct OT Map**. In this regard, Thm 3.1 plays a key role in identifying the sufficient condition under which the SNOT model avoids fake solutions.
>
> Moreover, while responding to the Reviewer YbFr, we were able to characterize additional cases where the SNOT model avoids fake solutions. **As an addtional contribution, we incorporated these additional results into the appendix as the following proposition**:
>
> > **Prop A.7** Suppose $\mathcal{X},\mathcal{Y}\subset M =\mathbb{R}^d$ satisfy Assumption A0. Then, the max-min solution of the SNOT problem recovers the correct OT map if any of the following conditions hold:
> >- $c$ is $C^1$, strictly convex with bounded Hessian, and $\mu$ does not assign positive mass to sets of Hausdorff dimension $d-1$.
> >- $\mathcal{X}$ and $\mathcal{Y}$ are compact, $\mu$ is absolutely continuous, and $c$ is a $C^1$ strictly convex.
> >- $c(x,y) = h(x-y)$ is a $C^1$ strictly convex and locally semi-concave function, where $h$ is a radially symmetric and strictly increasing function w.r.t $\Vert x-y\Vert$, and $\mu$ does not assign positive mass to sets of Hausdorff dimension $d-1$.
>
> $ $
>
> ---
> ### **Questions For Authors:**
>
> $ $
>
> > **Q.** Could you comment on the fact that the uniqueness in Thm 3.1. is in law, is this a limitation ? Especially, is it the same definition of uniqueness that is used in example 1, 2 ?
>
> **A.** Thank you for the thoughtful question. To avoid confusion, we revised the statement in Thm 3.1. as follows:
>
> > In particular, a map $x \mapsto y_{x} \in \cal{D}_{x}$ is a unique OT Map $T^{\star}$ $\mu$-a.s..
>
> **This is not a limitation, but rather a natural property of the OT Map.** The uniqueness of the OT Map $T^{\star}$ is always defined up to $\mu$-a.s., because modifying $T^{\star}$ on a set of measure zero does not affect the transport cost (Eq. 3). Specifically, this is the same definition of uniqueness that is used in example 1, 2.
>
> $ $
>
> ---
> > **Q.** For the second row of figure one, with what algorithm do you compute the max-min solution ? Are the results stochastic ?
>
> **A.** **The second row of Fig 1 presents manually visualized examples**, based on the analytical form of the max-min solution in Sec 3.2, rather than results by a numerical algorithm. In contrast, the first row of Fig 3 shows empirical results obtained by training the OTM model.
>
> $ $
>
> ---
> ### **Other Comments Or Suggestions:**
>
> $ $
>
> > **Q.** Citations, Typos, and Presentations
>
> **A.** We sincerely thank the reviewer for this detailed feedback. All comments regarding citations, typos, and presentations have been addressed as suggested.

---

### Official Review · Reviewer_YbFr · 2025-03-25

**Overall Recommendation:** 4

**Summary:**

This paper investigates convergence issues that arise when learning Optimal Transport (OT) maps using the Semi-Dual formulation. Specifically, the authors:
- Identify a sufficient condition under which the standard Semi-dual Neural OT approach converges to the correct OT map, resolving issues with inaccurate ("fake") solutions.
- Propose a method, OTP, which jointly learns both the OT map and/or the optimal transport plan, overcoming "fake" solutions.
- Provide theoretical guarantees showing that, under precise assumptions, OTP correctly solves the OT problem.
- Demonstrate experimentally that OTP successfully recovers OT maps where previous methods fail, on synthetic and image-to-image translation tasks.

**Claims And Evidence:**

All experimental claims are backed up with experiments on synthetic and real data. See more details in the Experimental Designs Or Analyses section.

**Essential References Not Discussed:**

N/A.

**Experimental Designs Or Analyses:**

- The experiments on synthetic data clearly illustrate and motivate the problem. However, they currently (i) rely on rather pathological cases and (ii) do not explore dimensions beyond $d=64$. I would suggest evaluating the proposed method on a benchmark for which the OT map is known in closed form, such as the one proposed in [Korotin+2022], where the OT map is given explicitly as the gradient of an ICNN. Additionally, this benchmark covers dimensions up to $d=256$. Given that the authors advocate for improved OT solutions, this additional experiment would clearly assess this claim, which in my opinion is not yet fully supported.

- To measure performance on image-to-image translation, the authors report FID and LPIPS. While I understand that FID (to some extent) measures the fitting of the marginal constraint $T\sharp\mu=\nu$, it is unclear to me why LPIPS can be interpreted as a measure of optimal transport cost. I suggest explicitly adding the following quantity as a metric to Table 2:
$$
\mathcal{C}(T) = \int_{\mathcal{X}} c(x, T(x)) d\mu(x)
$$
to properly investigate whether the proposed method achieves better transport-cost optimality.

**Methods And Evaluation Criteria:**

See the Experimental Designs Or Analyses section.

**Other Comments Or Suggestions:**

-  I would suggest explicitly clarifying this point in the abstract. Indeed, stating that the method "often generates fake solutions that fail to transfer one distribution to another accurately" (L18–19) might imply that the issue arises solely from violating the marginal constraint $T\sharp\mu=\nu$. However, as the authors correctly point out, fake solutions can potentially both violate the marginal constraint and fail to be cost-optimal, which is an important distinction to highlight.

**Other Strengths And Weaknesses:**

**Strengths:**

The paper proposes a very interesting idea of learning a sequence of transport map between sequences of noise-perturbed source and target distribution, with decreasing noise levels, akin to a diffusion models. They propose two smoothing scheme, and relate them VP and VE diffusion processes, making the method interpretable as a kind of diffusion model. Additionally, as the method eventually learn a "one-shot" single map between the source and the target distribution, the method could be interpreted as learning a learning a diffusion model and a "one-step" model distillation of it simultaneously during training, which is very interesting. For these reasons, I truly that this paper present a nice contribution that would be relevant for the conference. Yet, I have some concerns that I think should be fully in favor of acceptance.

**Weaknesses:**

***Min-Max optimization.***

A major weakness of the proposed methodology is its reliance on adversarial training, which is known to be challenging. This difficulty has notably motivated the recent preference for alternative approaches, such as diffusion models or flow/bridge matching frameworks.

***Theoretical Framework***

I have mixed feelings regarding the theoretical framework presented in this paper. In my view, the statements in the main text appear somewhat oversold. However, upon examining the appendix, it becomes clear that the authors have significantly simplified their results for presentation in the main text, and the motivation behind this simplification is unclear to me.

For example, in the main text, all the theoretical results are stated explicitly for the squared Euclidean cost $c(x, y) = \alpha \|x - y\|^2 $ with measures $\mu$ and $\nu$ supported on $\mathbb{R}^d$. More concretely, Theorem3.2 (L135--138) and Corollary~3.3 (L142--147) seem essentially to be straightforward reformulations of Brenier's theorem restricted to a compact subset of $\mathbb{R}^d$. Additionally, could the authors clarify why the assumption that the target measure has a density is required here? I may be misunderstanding this point, so some explanations would be appreciated.

On the other hand, in the appendix, the authors introduce a more general theoretical framework: measures are supported on a closed subset of a smooth, complete, connected Riemannian manifold, and the cost function is considered within a broader family of costs, including the squared geodesic distance. Given this broader framework, I am also curious why the authors did not explicitly state their results for more general cost functions of the form $c(x, y) = h(x - y)$ with $h$ strictly convex. Indeed, as mentioned in AppendixA.2 (L635--650), the Gangbo--McCann theorem guarantees the existence and uniqueness of the Monge map under this more general class of costs, provided that the source measure $\mu$ has a density. It thus appears that the main text results could naturally be generalized to this broader family of cost functions. Alternatively, if the authors specifically prefer the squared Euclidean cost $c(x, y) = \alpha \|x - y\|^2$, it might have been clearer to explicitly replace all generic cost references directly with this squared Euclidean form throughout the main text. Could the authors comment on this choice?

Finally, while it is indeed interesting that the authors introduce a weaker assumption---requiring only that the source distribution does not give mass to measurable sets of Hausdorff dimension at most $d - 1$---it seems that, in practice, their approach ultimately relies on the stronger condition that the source distribution has a density. This stronger condition is precisely what is used in the experiments, where the source measure is smoothed by introducing decreasing noise levels (whether in the VP or VE SDE framework), ensuring that the perturbed measure has a density. To fully appreciate the generality of their theoretical framework, it would have been interesting if the authors had also proposed perturbation schemes of the source measure satisfying their weaker condition, without necessarily imposing the stricter requirement of generating a density. Could the authors provide some insights into this decision as well?

***Evaluation.***  See the Experimental Designs Or Analyses section.

I would be more than willing to increase my score if the authors address these concerns. See the questions section.

**Questions For Authors:**

- Could the authors clarify why their choice of deriving general theory is presented in the appendix, yet restricted specifically to the squared Euclidean cost and measure on $\mathbb{R}^d$ for the main text?

- Could the authors expand upon the relationship between the LPIPS metric and the transport cost? It's unclear how LPIPS can be naturally interpreted as a transportation cost.

- It would be beneficial if the authors reported the transport cost for the image-to-image experiments, in addition to the LPIPS metric. This would allow for clearly distinguishing the source of experimental gains, determining whether improvements come from better satisfaction of the marginal constraint $T\sharp\mu=\nu$ (which is measured by FID), or from achieving better optimality through minimizing the transport cost itself.

- Could the authors apply their method to the benchmark introduced in [Korotin et al., 2022]? While the experiments in Table 1 provide convincing motivational examples, the constructed measures correspond to rather pathological cases and may not adequately demonstrate learning capabilities. Moreover, the current experiments only explore dimensions up to $d=64$, whereas this benchmark extends up to $d=256$, offering a more thorough evaluation.

- Lastly, could the authors elaborate on the connection between their approach and methods based on bridge matching?

I would be more than willing to increase my score if the authors address these concerns.

**Relation To Broader Scientific Literature:**

The author did a thorough background section on the literature about learning OT maps through the Semi-Dual OT formulation in Section 2, and notably adding more details in Section D of the Appendix. Yet, in my opinion, they lack relationship to other neural OT solvers, that are especially non-min-max optimization based.

First, while they include DSBM [Shi+2024] as a baseline in the experiments, they do not link their approach to bridge matching based neural OT solvers, such as DSBM [Shi+2024] itself, or [De Bortoli+2024], [Liu+2023], and flow matching based neural solvers, such as [Pooladian+2024], [Tong+2024:a,b], [Klein+2024]. In my opinion, this is a problem as I think that the the experimental gains observed are mainly due to the iterative refinement procedure, where we learn a map between a sequence of smoothed distributions. Especially, as the author advocate for learning stochastic OT maps, i.e. couplings, the bridge matching methods are particularly relevant.

Second, the author do not mention that some recent neural OT solvers [Uscidda+2023],  [Kassraie+2024] based relying on non-minimax objective functions that can approximate the OT maps as well and showed strong performances on benchmarks [Korotin+2022] where the OT map is known in closed form as the gradient of an ICNN. [Kassraie+2024] is of particular relevance to this work, as it learns a transport map between two distributions by solving a sequence of OT problems with decreasing entropic regularization.

[Shi+2024] Shi, Y., De Bortoli, V., Campbell, A., and Doucet, A. Diffusion schrodinger bridge matching. Advances in Neural Information Processing Systems, 36, 2024.

[Uscidda+2023] Théo Uscidda and Marco Cuturi. The monge gap: A regularizer to learn all transport maps, 2023.

[Tong+2024;a]  Alexander Tong, Nikolay Malkin, Guillaume Huguet, Yanlei Zhang, Jarrid Rector-Brooks, Kilian Fatras, Guy Wolf, and Yoshua Bengio. Conditional flow matching: Simulation-free dynamic optimal transport, 2024.

[Tong+2024;b] Alexander Tong, Nikolay Malkin, Kilian Fatras, Lazar Atanackovic, Yanlei Zhang, Guillaume
Huguet, Guy Wolf, and Yoshua Bengio. Simulation-free schrodinger bridges via score and flow matching.

[Kassraie+2024] P Kassraie, AA Pooladian, M Klein, J Thornton, J Niles-Weed, M Cuturi. Progressive entropic optimal transport solvers.

**Theoretical Claims:**

All theoretical claims are clearly supported by rigorous and, to my knowledge, correct proofs. Moreover, any informal assumptions or limitations of the theory are explicitly acknowledged, including:

> Proposition 3.4 (L263-266)
> *While the convergence theorem only guarantees convergence up to a subsequence (Thm. 4.1), our method exhibits decent convergence to $\pi^\star$ in practice (Sec. 5).* (L321-324)

---

> ### Author Rebuttal · Authors · 2025-03-31
>
> We sincerely thank the reviewer for carefully reading our manuscript and providing valuable feedback. We greatly appreciate the reviewer's acknowledgment that "this paper presents a nice contribution that would be relevant for the conference". We hope our responses to be helpful in addressing the reviewer's concerns.
>
> $ $
>
> ---
> ### **Questions For Authors:**
>
> $ $
>
> >**Q.** General theory in the appendix but quadratic cost in the main text?
>
> **A.** We appreciate the reviewer for providing helpful suggestions to improve our paper. In the original manuscript, we chose to focus on the quadratic cost function in the main text to **maintain consistency with our failure case analysis and experimental results**, which are all based on the squared Euclidean cost. Accordingly, in line 96, we defined $c$ as the quadratic function. Meanwhile, a more general setting is presented in Theorem A.4 in the appendix.
>
> Nevertheless, we agree with the reviewer that presenting a broader statement than Thm 3.1 would strengthen the paper. In this regard, instead of specifying a particular assumption in Thm 3.1, **we added the following proposition in the appendix that outlines additional settings under which Thm 3.1 holds** as follows:
> > **Proposition A.7** Let $\mathcal{X}, \mathcal{Y} \subset M = \mathbb{R}^d$ satisfy Assumption A0. Suppose that any one of the following conditions holds:
> >- The cost function $c$ is $C^1$, strictly convex with bounded Hessian, and the source distribution $\mu$ does not assign positive mass to sets of Hausdorff dimension $d-1$;
> >- The sets $\mathcal{X}$ and $\mathcal{Y}$ are compact, $\mu$ is absolutely continuous, and $c$ is a $C^1$ strictly convex function;
> >- The cost function $c(x,y) = h(x-y)$ is a $C^1$ strictly convex and locally semi-concave function, where $h$ is a radially symmetric and strictly increasing function with respect to $\Vert x-y\Vert$, and $\mu$ does not assign positive mass to sets of Hausdorff dimension $d-1$.
> >
> > Then, the max-min solution of the SNOT problem recovers the correct OT map.
>
> $ $
>
> ---
> >**Q.** Relationship between the LPIPS metric and the transport cost.
>
> **A.** Thank you for the insightful comment. We interpret the **LPIPS metric as a proxy for the transport cost in the feature space**. Due to the rebuttal length constraint, a more detailed explanation is provided at the following anonymous link:
>
> https://github.com/anonymousicml25/Overcome_Fake_Solution_OTP/tree/main
>
> $ $
>
> ---
> > **Q.** Additional experiments on the pathological cases (Table 1) with $d=256$ and on the benchmark introduced in [Korotin et al., 2022].
>
> **A.** We appreciate the reviewer for providing an insightful comment. Following the reviewer's suggestion, we conducted two additional experiments: (1) benchmark experiments from [1] and (2) 256-dim version of the failure case experiments in Table 1. Due to the rebuttal length constraint, we provide the full results at the following anonymous link:
>
> https://github.com/anonymousicml25/Overcome_Fake_Solution_OTP/tree/main
>
> Reference
> - [1] Korotin et al., "Do Neural Optimal Transport Solvers Work? A Continuous Wasserstein-2 Benchmark.", NeurIPS, 2021.
>
> $ $
>
> ---
> > **Q.** Additional related work: Bridge matching and non-minimax approaches
>
> **A.** We appreciate the reviewer for suggesting these relevant works. We incorporated additional related works, such as bridge matching approaches and non-minimax approaches in Appendix D.
>
> $ $
>
> ---
> ### **Other Strength and Weakness:**
>
> $ $
>
> > **Q.** Thm 3.2 and Cor 3.3 seem essentially to be straightforward reformulations of Brenier's theorem. Additionally, why the assumption that the target measure has a density is required here?
>
> **A.** We agree with the reviewer that Thm 3.2 and Cor 3.3 can be derived from classical results such as Brenier’s theorem. As noted in Lines 125–129, we included these results for the sake of **completeness within the context of SNOT problems**. In this regard, we would like to emphasize that, to the best of our knowledge, Cor 3.3 is the first result to explicitly state a sufficient condition for the **uniqueness of the max-min solution in the SNOT objective**.
>
> Regarding the assumption that the target measure has a positive density on its domain $\mathcal{Y}$, **this condition ensures that the Kantorovich potential is unique (up to a constant) within $\mathcal{Y}$**. If the target distribution has disconnected support, the potential can differ by locally constant functions across its connected components.
>
> $ $
>
> ---
> ### **Other Comments Or Suggestions:**
>
> $ $
>
> >**Q.** Fake solutions may violate the marginal constraint or cost optimality.
>
> **A.** Thank you for the great question. Interestingly, **any max-min solution satisfying the marginal constraint is guaranteed to be the optimal transport map**. Due to the rebuttal length constraint, a more detailed explanation is provided at the following anonymous link:
>
> https://github.com/anonymousicml25/Overcome_Fake_Solution_OTP/tree/main

---

> > ### Comment · Reviewer_YbFr · 2025-04-01
> >
> > I thank the authors for their thorough answers to my concerns. I am happy to raise my score form 3 to 4.

---

> > > ### Author Response · Authors · 2025-04-04
> > >
> > > We are happy to know that the most of the reviewer’s concerns have been addressed. We also appreciate your valuable feedback and will incorporate the modifications and new results in the final version of the manuscript.

---

### Decision · Program_Chairs · 2025-05-01

**Decision:**

Accept (poster)

**Comment:**

The paper presents a thoughtful and well-motivated approach to overcoming known issues in semi-dual neural optimal transport, specifically addressing the occurrence of spurious solutions. The proposed method, OTP, introduces a smoothing-based strategy with solid theoretical backing, including a sufficient condition for convergence to the correct transport map. The paper also delivers practical insights through a sequence of carefully designed experiments, demonstrating improved stability and performance compared to standard baselines in both synthetic and real-world tasks. The reviewers highlighted several strengths: clear identification of a key limitation in existing semi-dual OT methods, meaningful theoretical contributions (e.g., the condition on Hausdorff dimension), and the design of a simple yet effective regularization scheme via noise scheduling. The authors’ responses during the rebuttal were thorough and constructive. They expanded the theoretical scope of their results, clarified design choices, conducted additional experiments on established benchmarks, and addressed reviewer suggestions regarding terminology and related work. Overall, this is a technically sound, well-presented, and timely contribution that advances our understanding and practical handling of neural OT models. I recommend acceptance.